



Version of May 22, 2023

# Evaluation of vertical transport in the Asian monsoon 2017 from CO$_2$ reconstruction in the ERA5 and ERA-Interim reanalysis

Bärbel Vogel[1], Michael Volk[2], Johannes Wintel[2,*], Valentin Lauther[2], Jan Clemens[1,3], Jens-Uwe Grooß[1], Gebhard Günther[1], Lars Hoffmann[3], Johannes C. Laube[1], Rolf Müller[1], Felix Ploeger[1], and Fred Stroh[1]

[1]Institute of Energy and Climate Research (IEK-7), Forschungszentrum Jülich, Jülich, Germany
[2]Institute for Atmospheric and Environmental Research, University of Wuppertal, Wuppertal, Germany
[3]Jülich Supercomputing Centre, Forschungszentrum Jülich, Jülich, Germany
[*]now at: Curt-Engelhorn-Centre of Archaeometry gGmbH, Mannheim, Germany

**Correspondence:** B. Vogel (b.vogel@fz-juelich.de)

**Abstract.**

Atmospheric concentrations of many greenhouse gases especially CO$_2$ are increasing globally. In particular the rapid increase of anthropogenic CO$_2$ emissions in Asia contributes strongly to the acceleration of the CO$_2$ growth rate in the atmo-

sphere. During the Asian monsoon season, greenhouse gases as well as pollution emitted near the ground rapidly propagate up to an altitude of 13 km ($\sim$360 K potential temperature) with slower ascent and mixing with the stratospheric background above. However, CO$_2$ sources in South Asia are poorly quantified. Here, differences in transport of air in the regions of the Asian summer monsoon 2017 were inferred using the Chemical Lagrangian Model of the Stratosphere (CLaMS) driven by three data sets, namely two ECMWF reanalyses in different resolutions (ERA-Interim, ERA5 and ERA5 $1° \times 1°$). These model results

are assessed using unique airborne measurements up to altitudes of $\sim$20 km ($\sim$475 K) during the Asian summer monsoon 2017 conducted with the Geophysica aircraft during the StratoClim campaign in Nepal. Trajectory-based transport times, air mass source regions at the Earth's surface, mean effective ascent rates and age spectra as well as mean age of air from 3-dimensional CLaMS simulations are compared using the three data sets and evaluated by observation-based ascent rates. Our findings confirm that because of a better spatial and temporal resolution, ERA5 reanalysis yields a better representation of convection than

ERA-Interim. Further, our findings show that transport times from the surface to the Asian monsoon anticyclone as well as the origin of air at the Earth's surface are both very sensitive to the used reanalysis. Above 430 K, the mean effective ascent rates derived from ERA5 back-trajectories and ERA5 $1° \times 1°$ ($\approx$ 0.2-0.3 K/day) are in good agreement with the observation-based mean ascent rates inferred from long-lived trace gases such as C$_2$F$_6$ and HFC-125 derived from air samples collected by the whole air sampler aboard Geophysica. Mean effective ascent rates derived from ERA-Interim back-trajectories are much faster

$\approx$ 0.5 K/day at these altitudes. In the Asian monsoon region at 470 K, mean age of air is larger than 3 years for ERA5 $1° \times 1°$ and about 2 years for ERA-Interim, whereas an observation-based age of air is up to 2.5 years.

A reliable reconstruction (simulation) of vertical CO$_2$ profiles during the Asian monsoon is a challenge for model simulations because the seasonal variability of CO$_2$ at the ground, mixing with aged stratospheric air and the vertical velocities (including convection as well as vertical ascent caused by diabatic heating in the UTLS) have to be represented accurately in





the simulations. Up to 410 K, the presented $CO_2$ reconstruction agrees best with high-resolution in situ aircraft $CO_2$ measurements using ERA5 compared to ERA5 $1° \times 1°$ and ERA-Interim, indicating a better representation of Asian monsoon transport for the newer ECMWF reanalysis product ERA5. Above 410 K the uncertainties of the $CO_2$ reconstruction are increasing because of mixing with aged air.

## 1  Introduction

The global amount of greenhouse gases (GHGs) in the atmosphere has increased because of worldwide anthropogenic emissions. In particular, the rapid increase of anthropogenic $CO_2$ emissions in South Asia contributes strongly to the acceleration of the $CO_2$ growth rate, e.g. the anthropogenic $CO_2$ emission rate from India was the fourth highest worldwide in 2017 (behind China, the USA and the European Union) (Friedlingstein et al., 2019, 2022). In addition to GHGs also pollution, water vapour, aerosols and their precursors as well as some ozone-destroying substances have high emission rates in Asia and can

be transported very fast into the lower stratosphere during the Asian summer monsoon season (e.g. Brunamonti et al., 2018; Hanumanthu et al., 2020; Adcock et al., 2021; Appel et al., 2022; Vogel et al., 2023). Subsequently these trace gases as well as aerosol can be distributed into the global northern lower stratosphere over a period of several weeks (e.g. Ploeger et al., 2013; Müller et al., 2016; Vogel et al., 2016; Yu et al., 2017; Rolf et al., 2018; Lauther et al., 2022). To better understand the impact of anthropogenic emissions in Asia on the atmosphere, it is important to evaluate the transport of air in the Asian summer mon-

soon region into the lower stratosphere represented in meteorological reanalyses in combination with unique high-resolutions aircraft measurements obtained over the northern Indian subcontinent during the StratoClim aircraft campaign in summer 2017 (Stroh and StratoClim-Team, 2023).

From about June to September, the Asian summer monsoon constitutes a seasonally persistent zonally restricted circulation pattern transporting climate-relevant emissions rapidly from surface sources to higher altitudes, i.e. to the lower stratosphere

(e.g. Mason and Anderson, 1963; Randel and Park, 2006; Park et al., 2007; Pan et al., 2016; Vogel et al., 2015; Ploeger et al., 2017; Vogel et al., 2023). The Asian summer monsoon is associated with deep convection over the Indian subcontinent and an anticyclonic flow in the upper troposphere and lower stratosphere (UTLS) over the Asian monsoon region spanning from northeast Africa to the Pacific (e.g. Park et al., 2007). Air parcels are uplifted quickly by convection followed by slow diabatic uplift in the UTLS superimposed by the anticyclonic flow (e.g. Brunamonti et al., 2018; Vogel et al., 2019; Legras and Bucci,

2020; von Hobe et al., 2021), while in other regions within the tropical transition layer the heating rates are in general smaller during boreal summer (Vogel et al., 2019). The higher the air parcels are located above the level of maximum convective outflow ($\sim 360$ K, $\sim 13$ km), the larger the contribution of air masses is from outside the Asian monsoon anticyclone (i.e. from the stratospheric background) to the upward spiralling flow (Vogel et al., 2019, 2023).

In state-of-the-art chemistry transport models, the transport of air parcels differs, because different methods (Eulerian, La-

grangian), different vertical velocities (kinematic, diabatic) and different meteorological reanalyses are used to drive the models (e.g. Stenke et al., 2009; Bergman et al., 2013; Brinkop and Jöckel, 2019; Tao et al., 2019; Ploeger et al., 2019; Bucci et al., 2020; Legras and Bucci, 2020; Tegtmeier et al., 2020). Further, the implementation of convection and irreversible mixing dif-



fers from model to model (e.g. Konopka et al., 2019; Wohltmann et al., 2019; Hoffmann et al., 2023). The aim of our study is to infer differences of vertical transport in the Asian monsoon region using three data sets: two reanalyses provided by the European Centre for Medium-Range Weather Forecasts (ECMWF), namely, ERA-Interim and its successor ERA5 as well as its down-scaled version ERA5 $1° \times 1°$, a computing-time-saving alternative to the full resolution ERA5 data.

In general, differences between ERA-interim and ERA5 are attributed to the better spatial and temporal resolution of the ERA5 reanalysis, which allows for a better representation of convective updrafts, gravity waves and tropical cyclones (e.g. typhoons) (e.g. Hoffmann et al., 2019; Li et al., 2020; Legras and Bucci, 2020; Malakar et al., 2020). Consequently, ERA5 provides a more accurate representation of the lapse rate tropopause height than ERA-Interim (e.g. Hoffmann et al., 2022; Tegtmeier and Krüger, 2022).

In the Asian monsoon anticyclone slow diabatic uplift in the range of 1-1.5K per day was found above the level of maximum convective outflow using Lagrangian transport simulations driven by ERA-Interim (Vogel et al., 2019). However, it was found consistently in several previous studies that in general the vertical velocities in ERA-Interim are 30-50% too fast in the tropics (Dee et al., 2011; Ploeger et al., 2012; Schoeberl et al., 2012). Tegtmeier and Krüger (2022) summarise that diabatic vertical ascent appears to be faster in ERA-Interim, which produces a residence time (between 370 K and 400 K) of $\sim$2 months in the tropical tropopause layer, in contrast to residence times of $\sim$3 months or longer based on other reanalyses (e.g MERRA, MERRA-2, or CFSR, however ERA5 was not included here). This bias seems to be corrected in ERA5 manifesting in weaker diabatic heating rates in the tropics resulting in a higher age of air (i.e., larger mean stratospheric transit times) and thus a significantly slower Brewer–Dobson circulation in ERA5 compared to ERA-Interim (Ploeger et al., 2021). Different residence times in the UTLS would change the chemical composition in these altitudes and even small changes of radiative active trace gases such as $O_3$, $H_2O$ or aerosol could have important local radiative impacts (e.g. Riese et al., 2012; Vernier et al., 2015; Fadnavis et al., 2019; Bian et al., 2020).

For the StratoClim aircraft campaign during the Asian summer monsoon 2017, in general, a higher consistency with observed data and a better reproducibility of pollution features could be found using ERA5 compared to ERA-Interim using diabatic trajectory calculations back to cloud top altitudes (Bucci et al., 2020). ERA5 improves in general the transport in the Asian monsoon 2017, however upward transport in the region of the Tibetan Plateau, should be considered with caution (Legras and Bucci, 2020). Considering the transport of air masses contributing to the Asian Tropopause Aerosol Layer (ATAL) measured in the region of the Asian monsoon anticyclone during August 2016, ERA5 shows in general faster transport of air from the ground to ATAL altitudes (up to $\approx$410 K) due to a better representation of convection. In addition, more continental source regions contributing to the ATAL are found in ERA5, whereas in ERA-Interim more marine sources are attributed to air in ATAL altitudes (Clemens et al., 2023).

Different vertical velocities or ascent rates in the region of the Asian monsoon anticyclone have consequences for global transport simulations using ERA-Interim or ERA5 reanalysis. To assess such global simulations it is essential to understand the strengths and weaknesses of the newest ECMWF product ERA5 in particular in the Asian monsoon region. In this work, differences in the transport of air in the regions of the Asian summer monsoon 2017 will be inferred using the Chemical Lagrangian Model of the Stratosphere (CLaMS) driven by the three data sets (ERA-Interim, ERA5 and ERA5 $1° \times 1°$). Model



results will be assessed using unique airborne measurements up to $\sim$20 km during the Asian summer monsoon 2017 conducted during the StratoClim aircraft campaign in Nepal (Stroh and StratoClim-Team, 2023). Trajectory-based transport times, origin

of air at the Earth's surface, mean effective ascent rates and age spectra as well as mean age of air from 3-dimensional CLaMS simulations are compared using the three data sets (ERA-Interim, ERA5 and ERA5 $1° \times 1°$). In addition, age of air as well as mean effective ascent rates are compared to observation-based age of air and mean ascent rates inferred from long-lived trace gases such as $C_2F_6$ and HFC-125.

Further, a unique set of $CO_2$ aircraft measurements featuring high temporal and vertical resolution up to $\sim$20 km was

obtained during the StratoClim aircraft campaign 2017 (Stroh and StratoClim-Team, 2023). Measured $CO_2$ profiles are successfully reconstructed using ground-based measurements of $CO_2$ mainly from Nainital (northern India) by Lagrangian model simulations using ERA5 reanalysis leading to an improved understanding of the vertical structure of $CO_2$ in the monsoon region (Vogel et al., 2023). Here we use the same $CO_2$ reconstruction method as used in Vogel et al. (2023), but focus on the differences between the three data sets (ERA-Interim, ERA5 and ERA5 $1° \times 1°$). Further, trajectory-based $CO_2$ transport

times are evaluated using estimated mean transport times from ground-based $CO_2$ measurements. In general, our results show that using ERA5 reanalysis will yield a better agreement with aircraft measurements conducted over the Indian subcontinent in summer 2017 as ERA-Interim.

## 2 Measurements during the Asian summer monsoon 2017

In the frame of the StratoClim project funded by the European Commission, a measurement campaign using the Russian Geo-

physica high altitude research aircraft was conducted in Kathmandu (Nepal) in summer 2017 (see Fig. 1) to measure a variety of trace gases and aerosol characteristics for the first time in the Asian monsoon anticyclone up to 20 km altitude (corresponding to $\sim$55 hPa or $\sim$475 K potential temperature) (Stroh and StratoClim-Team, 2023). The StratoClim measurements constitute a unique data set to characterise major processes which dominate particle and trace gas transport from the northern Indian subcontinent, one of the most polluted regions of the world, into the lower stratosphere.

$CO_2$ was detected using the multi-tracer in situ instrument HAGAR (Volk et al., 2000; Homan et al., 2010) operated by the University of Wuppertal. Apart from $CO_2$, it also provides simultaneous in situ measurements of $N_2O$, $CH_4$, CFC-12, CFC-11, H-1211, $SF_6$ and $H_2$. Except for $CO_2$, which is measured at high time resolution (3 to 5 s) by non-dispersive infrared absorption (NDIR), all the other species were measured by gas chromatography with electron capture detection (GC/ECD) every 90 s. The instrument is calibrated every 7.5 min during flight with either of two standard gases, which are inter-calibrated

in the laboratory with standards provided by NOAA GML. For StratoClim the accuracy of $CO_2$ was estimated to be about 0.2 ppm. HAGAR $CO_2$ was referenced to standards provided by NOAA and are based on the WMO X2007 scale and can be converted to the current WMO X2019 (using X2019 = 1.00033 * X2007 + 0.0467) based on reassigned standard values on the current scale.

The long-lived trace gases $C_2F_6$ and HFC-125 were collected with the whole air sampler of Utrecht University operated on

board the Geophysica research aircraft (e.g. Laube et al., 2010a). Ambient air was compressed into evacuated stainless-steel



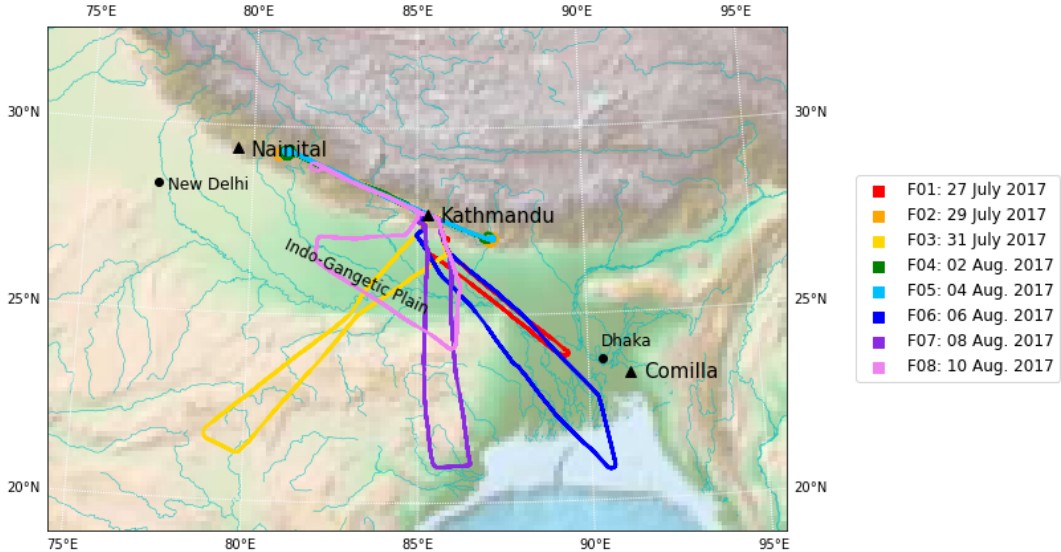

**Figure 1.** Regional map of the aircraft measurements over the Indian subcontinent. The flight paths of the eight local scientific flights (F01-F08) by the high altitude research aircraft Geophysica are shown. The scientific flights were carried out every second day from Kathmandu (Nepal) between 27 July and 10 August 2017. In addition, the locations of the measurement sites for Greenhouse gases in Nainital (NTL, India) and Comilla (CLA, Bangladesh) used for $CO_2$ reconstruction are indicated (figure adapted from Vogel et al. (2023)).

canisters (2 L) using a metal bellows pump that has been previously shown to not impact trace gas mixing ratios (e.g. Kaiser et al., 2006). The samples were transported to the University of East Anglia (UEA) for analysis on a high sensitivity gas chromatograph-trisector mass spectrometer system (Laube et al., 2010b). More details on the whole air sampler measurements during StratoClim and the used analytical technique can be found in Adcock et al. (2021).


# 3 Lagrangian transport simulations

## 3.1 CLaMS trajectory calculations

Trajectory calculations were performed using the Chemical Lagrangian Model of the Stratosphere (CLaMS) (McKenna et al., 2002b, a; Pommrich et al., 2014, and references therein) which was developed with the aim to study transport and chemi-
cal processes throughout the troposphere and stratosphere in the presence of strong tracer gradients. Here, CLaMS diabatic backward trajectories were started along the entire flight paths (every 1 second) of all Geophysica flights conducted over the northeastern part of the Indian subcontinent. Overall ∼110000 back-trajectories are calculated, between 9000 and 16000 per flight depending on the flight lengths.





For comparison, the back-trajectory calculations are driven by three data sets, two reanalyses in different resolutions provided
by the European Centre for Medium-Range Weather Forecasts (ECMWF): ERA-Interim, ERA5 and ERA5 $1° \times 1°$. The new
ERA5 reanalysis (Hersbach et al., 2020) is a high-resolution atmospheric data set with 137 vertical levels up to 0.01 hPa, a
horizontal resolution of $\sim 31$ km ($T_L$ 639) and a hourly time resolution. We retrieved the data on a $0.3° \times 0.3°$ horizontal grid.
The ECMWF's prior reanalysis ERA-Interim (Dee et al., 2011) has 60 vertical levels up to 0.01 hPa, a horizontal resolution of
$\sim 79$ km ($T_L$ 255) (corresponding to $1° \times 1°$ horizontal grid) and a 6-hourly time resolution. In addition, we use ERA5 data in
lower resolution referred to as 'ERA5 $1° \times 1°$' (similar to Ploeger et al., 2021; Konopka et al., 2022). For ERA5 $1° \times 1°$, ERA5
data are truncated to a $1° \times 1°$ horizontal grid and a 6-hourly time resolution (same as ERA-Interim). The vertical resolution is
the same as in the original ERA5 reanalysis. ERA5 $1° \times 1°$ data are a computing-time-saving alternative to the full resolution
ERA5 data and are particular suited for 3-dimensional global multi-annual CLaMS simulations.

In the CLaMS model, potential temperature is used as the vertical coordinate when the pressure is less than about 300 hPa,
(i.e. in the upper troposphere and in the stratosphere); when the pressure is greater than about 300 hPa (more accurately, for
pressure p exceeding a reference level of p/p$_{\text{surface}}$ = 0.3), a pressure-based orography-following hybrid coordinate (in units
of K) is used (Pommrich et al., 2014). In potential temperature levels above about 300 hPa, the vertical velocity is determined
solely by the total heating rate (Pommrich et al., 2014; Ploeger et al., 2021). Total diabatic heating rates include clear-sky
radiative heating, cloud radiation, latent heat release, as well as turbulent and diffusive heat transport for the upper troposphere
and stratosphere are used from ECMWF reanalyses.

Trajectories are considered ending in the model boundary layer referred to as 'model BL', when they are located for the
first time below about 2–3 km above surface considering orography (i.e., the vertical hybrid pressure–potential–temperature
coordinate ($\zeta$) fulfils $\zeta \leq 120$ K) (details see e.g. Vogel et al., 2015, 2019).

### 3.2 Method for $CO_2$ reconstruction

Vogel et al. (2023) demonstrated that a reasonable reconstruction of vertical $CO_2$ profiles measured during the StratoClim
campaign in Nepal during summer 2017 can be conducted successfully using CLaMS back-trajectories driven by ERA5 re-
analysis. Following the approach by Vogel et al. (2023), here we apply the same method for $CO_2$ reconstruction, however
the differences between ERA5 compared to ERA-Interim and ERA5 $1° \times 1°$ will be analysed to infer possible differences in
transport of air masses between the three data sets.

The method for $CO_2$ reconstruction used in Vogel et al. (2023) is briefly summarised hereafter. $CO_2$ mixing ratios from
ground-based observations on the Indian subcontinent (Fig. 2) measured during the time when the CLaMS back-trajectories
reach the model BL are used for $CO_2$ reconstruction. For that purpose different ground-based observations available on differ-
ent time scales (monthly, weekly or daily) were interpolated in time on a daily grid. These calculated $CO_2$ mixing ratios define
$CO_2$ in the model boundary layer and are transported passively along the trajectory to the location and time of the Geophysica
flight path, i.e. $CO_2$ is treated as chemically inert over the time of the back-trajectory calculation.

As a second step, a regional mask was developed where $CO_2$ is prescribed in the model BL depending on different geo-
graphical regions (see Fig. 3). In each of these geographical regions referred to as 'BL region' $CO_2$ is prescribed using one





specific measurement site, e.g. trajectories ending in the BL region marked in green and dark-red (roughly Indian Subcontinent and Tibetan Plateau) are prescribed using ground-based measurements from Nainital and the BL region marked in yellow
(roughly Bangladesh) is prescribed using ground-based measurements from Comilla. Unfortunately the coverage of ground-based measurements of $CO_2$ over the Indian subcontinent in 2016 to 2017 is sparse, therefore only data from Nainital and Comilla are available. Additional $CO_2$ ground-based time series from other geographical regions influencing the Geophysica measurements provided by measurement sites for Greenhouse gases in Mt. Waliguan (WLG, China), Bukit Kototabang (BKT, Indonesia), Mauna Loa (MLO, Hawaii) and Samoa (SMO, Cape Matatula) are used for $CO_2$ reconstruction (more details about
the used ground-based observations can be found in Vogel et al., 2023).

The seasonal variability of $CO_2$ over the northern Indian subcontinent (mean value between 20–30°N and 75–95°E) of the lowest model level at 975 hPa of the GOSAT-L4B product (Matsunaga, T. and Maksyutov, S. (eds.), 2018) is shown in Fig. 2 for comparison to ground-based $CO_2$ measurements. The GOSAT-L4B product is a model simulation using $CO_2$ surface fluxes inferred from column-averaged satellite measurements (Maksyutov et al., 2013). The lowest model level of GOSAT-L4B is
closest to the inferred $CO_2$ surface fluxes and is not strongly influenced by the tracer transport of the underlying transport model. GOSAT-L4B $CO_2$ over the northern Indian subcontinent has the same seasonality as ground-based $CO_2$ over Nainital, however the minimum and maximum values differ strongly, highlighting the need of ground-based $CO_2$ measurements over the Indian subcontinent in addition to satellite-based estimation of $CO_2$ surface fluxes (a more detailed discussion can be found in Vogel et al., 2023).


### 3.3 Mean age of air from 3-dimensional CLaMS simulations

Trajectory-based transport times are compared to mean age of air from global 3-dimensional CLaMS chemistry transport model simulations performed over a time period of several decades to consider in addition the transport time of aged air, that is not considered in our pure back-trajectory calculations ending on 1 June 2016. Global 3-dimensional CLaMS simulations
are based on 3-dimensional forward trajectories and a parametrisation of small-scale mixing depending on the shear in the large-scale flow (e.g. Pommrich et al., 2014). The model simulations are driven with either ERA5 $1° \times 1°$ or ERA–Interim reanalysis winds and diabatic heating rates, as described in more detail by Ploeger et al. (2021).

Global 3-dimensional CLaMS simulations are used to calculate the age of air spectrum, the distribution of transit times through the stratosphere, at each location in the stratosphere based on chemically inert pulse tracers (e.g. Ploeger et al., 2021).
The 60 different tracer pulses are released at the tropical surface (30°S-30°N), more specifically by a mixing ratio boundary condition in the lowest model layer. The chosen pulse frequency of 2 months allows a 2–month resolution of the age spectrum along the transit time axis for 10 years of transit time (more details in Ploeger et al., 2021). Mean age of air is calculated in three different ways to enable assessing uncertainties arising from the method. First, mean age is calculated as the first moment (mean) of the age spectrum, Second, this age spectrum–based mean age has been corrected for its finite tail (truncated to 10
years), by applying an exponential correction fit. Third, mean age has also been calculated from a clock-tracer with a linear increase in the entire lowest model layer. Due to the methodological differences, the age spectrum–based mean age is expected





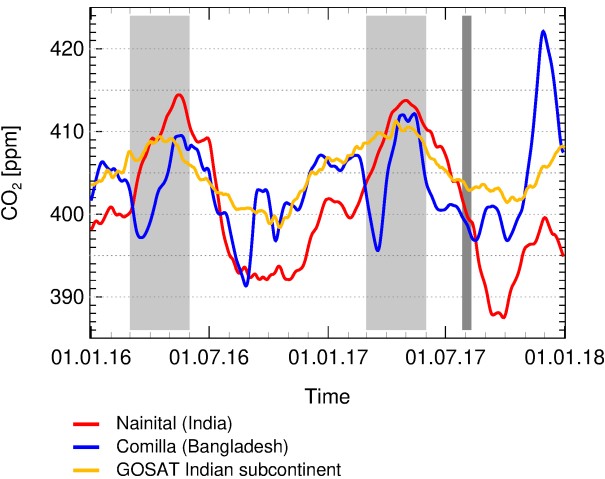

**Figure 2.** Temporal variability of ground-based $CO_2$. The variability of ground-based $CO_2$ is shown at Nainital and Comilla (geographical positions see Fig. 1). In addition, the seasonal variability of $CO_2$ over the northern Indian subcontinent (mean value between 20–30°N and 75–95°E) of the lowest model level at 975 hPa of the GOSAT-L4B product for comparison to ground-based $CO_2$ measurements is shown. The pre-monsoon period (March–May) when a seasonal $CO_2$ maximum is expected is high-lighted (light-grey) as well as the period of the StratoClim aircraft campaign during monsoon 2017 (dark-grey).

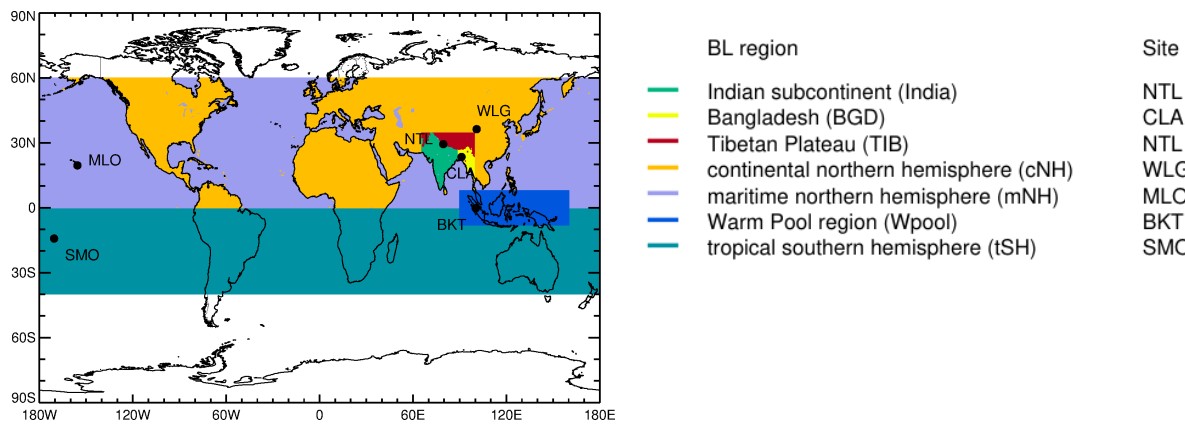

**Figure 3.** Regional mask for $CO_2$ reconstruction using $CO_2$ ground-based measurements at different sites in Asia and the Pacific. In each model boundary layer (BL) region (marked by different colours) $CO_2$ is prescribed from one specific measurement site: tropical southern hemisphere (tSH) by Samoa (SMO), Indian subcontinent (India) by Nainital (NTL), Bangladesh (BGD) by Comilla (CLA), Tibetan Plateau (TIB) by Nainital (NTL), marine northern hemisphere (mNH) by Mauna Loa (MLO), continental northern hemisphere (cNH) by Mt. Waliguan (WLG) and Warm Pool region (Wpool) by Bukit Kototabang (BKT) (figure adapted from Vogel et al. (2023)).





| season | time period | start time | age of air |
|---|---|---|---|
| monsoon 2017 | June–September 2017 | 1 June 2017 | ∼ 2 months |
| pre-monsoon 2017 | March–May 2017 | 1 March 2017 | ∼ 2–5 months |
| winter 16/17 | December 2016 – February 2017 | 1 Dec 2016 | ∼ 5–8 months |
| post-monsoon 2016 | October–November 2016 | 1 Oct 2016 | ∼ 8–10 months |
| monsoon 2016 | June–September 2016 | 1 June 2016 | ∼ 10–14 months |
| aged air | older than 1 June 2016 | | > 14 months |

**Table 1.** Time periods and trajectory-based age of air of considered seasons on Indian subcontinent. The analysis of CLaMS back-trajectories is performed back until the start time of each season. For each season, air parcels that were released at the model boundary layer (BL) are analysed. The longest simulation time is back until 1 June 2016 (∼ one year). Air parcels that are located in the free atmosphere on 1 June 2016 are considered as aged air.

to yield the youngest estimate, the spectrum-based mean age including the tail correction yields the oldest estimate, and the clock-tracer mean age values lie in between. The range between these three different mean age estimates can be interpreted as an estimate for methodological uncertainties arising from the mean age calculation. Compared to the trajectory-based transport

times, all three mean age estimates are expected to result in higher values, as they include the effects of mixing and recirculation of old stratospheric air into the tropics, which is absent in the pure trajectory calculations ending on 1 June 2016.

## 4 Results

CLaMS diabatic backward trajectories driven by three data sets (ERA-Interim, ERA5 and ERA5 $1° \times 1°$) were started along

the entire flight paths (every 1 second) of all Geophysica flights (F01-F08) performed over the Indian subcontinent to infer a trajectory-based transport time from the location of the measurement back to the time when the air parcel was released at the model boundary layer (BL). The trajectories are calculated back to 1 June 2016 and are analysed within different time periods to identify the source regions at the model BL depending on season (see Tab. 1). However, most air parcels were released at the model BL much later than 1 June 2017, e.g. 64% (63%) of all air parcels are from the monsoon season 2017 using ERA5

(ERA-Interim) reanalysis.

The higher the sampled air parcels are located the longer are their simulated trajectory-based transport times which is to be expected. However, there is also a strong variability of transport times between individual air parcels at the same level of potential temperature indicating mixing of air masses of different transport times (or different ages) as well as from different origins.

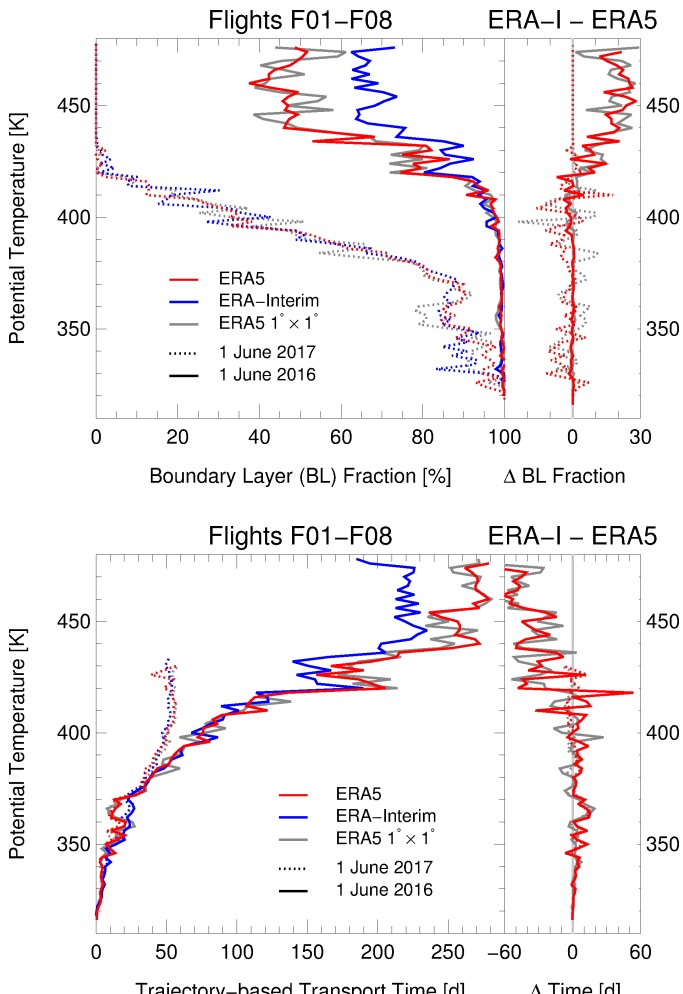

**Figure 4.** Mean fraction of air from the model BL (top) and the trajectory-based mean transport time (bottom) as calculated from all backward trajectories started along the Geophysica flight tracks averaged as median in 2 K intervals and accumulated back to the start times of monsoon 2017 (1 June 2017, dotted lines) and monsoon 2016 (1 June 2016, solid lines). The trajectory calculations are driven by three data sets (ERA-Interim, ERA5 and ERA5 $1° \times 1°$) indicated by different colours. The differences ($\Delta$) between ERA-Interim and ERA5 and ERA5 $1° \times 1°$, respectively, are shown in the right panels.




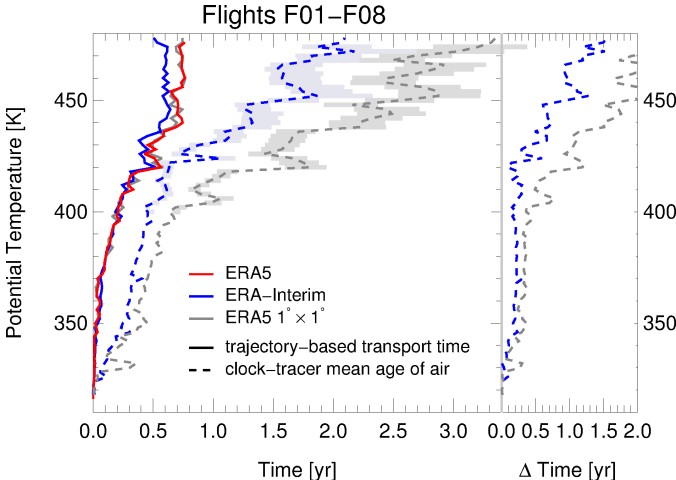

**Figure 5.** Trajectory-based mean transport time back to monsoon 2016 (same as in Fig. 4a) and the clock-tracer mean age of air inferred from 3-dimensional CLaMS simulations averaged as median in 2 K intervals. The trajectory calculations are driven by three data sets (ERA-Interim, ERA5 and ERA5 $1° \times 1°$). Mean age of air is only available for 3-dimensional CLaMS simulations driven by ERA-Interim and ERA5 $1° \times 1°$. The time differences ($\Delta$ Time) between clock-tracer mean age of air and trajectory-based mean transport time are shown in the right panel. Methodological differences in calculating mean age of air are indicated as shading, showing in addition the difference between the age spectrum–based mean age and the spectrum-based mean age including the tail correction (Sect. 3.3). Due to the methodological differences, the age spectrum–based mean age yields the youngest estimate, the spectrum-based mean age including the tail correction yields the oldest estimate, and the clock-tracer mean age values lie in between (below 400 K the difference between the three methods is minor and therefore not shown).

## 4.1 Transport times and mean age of air

The mean fraction of air from the model BL and the trajectory-based mean transport time are calculated from all backward trajectories started along the Geophysica flight tracks depending on flight height. Fig. 4 shows the trajectory-based mean transport time (averaged as median in 2 K intervals) and the mean fraction of air from the model BL depending on potential temperature and accumulated back to the start times of monsoon 2017 and monsoon 2016 using the three data sets. To calculate the trajectory-based mean transport time only the fraction of trajectories from the model BL is considered, thus older air masses are neglected as first approximation.

All simulations show that considering a trajectory length back to the start time of monsoon 2017 yield a boundary layer fraction between 80% and 100% below 370K; above the boundary layer fraction is decreasing rapidly and reaches 0% around 420 K (Fig. 4a). Thus a trajectory length of about two months is too short for a comprising simulation of the chemical composition of the Asian monsoon anticyclone because only very young air masses are considered.



Using a trajectory length back to the start time of monsoon 2016 ($\approx$ 10-14 months), a boundary layer fraction of almost 100% is reached up to 410 K. Above, the boundary layer fraction is slowly decreasing and depends strongly on the used ECMWF reanalysis and implicates different mean transport times at these altitudes (Fig. 4b).

Below 410 K convection in ERA5 yields faster transport times (up to $\approx$20 days) and higher model boundary layer (BL) fractions than in ERA-Interim. Vice versa, above 420 K air masses have faster transport times up to two months ($\approx$60 days) from the model BL to the UTLS in ERA-Interim compared to ERA5 corresponding to a 20% higher fraction from the model BL. Transport times inferred from ERA5 $1° \times 1°$ trajectories have in principle a similar behaviour as using ERA5, however the variability is somewhat different, caused by the different temporal and horizontal resolution.

Considering in addition aged air (older than 1 June 2016) the mean age of air from a 3-dimensional CLaMS simulation (Sect. 3.3) is compared to the trajectory-based mean transport times calculated from pure back-trajectory calculations (Fig. 5). Mean age of air is only available for 3-dimensional CLaMS simulations driven by ERA-Interim and ERA5 $1° \times 1°$. The trajectory-based mean transport times of ERA5 and ERA5 $1° \times 1°$ in the lower stratosphere are very similar as shown in Fig. 4b, therefore we assume that also the mean age of air from 3-dimensional CLaMS simulations driven by ERA5 and ERA5 $1° \times 1°$ would be similar.

$N_2O$ profiles measured during the StratoClim campaign indicate strong mixing with older stratospheric air above $\sim$400 K (Vogel et al., 2023). Halon-1211 (which has a shorter live-time as $N_2O$) measurements aboard Geophysica (see Fig. 2 in Adcock et al., 2021) indicate that just below $\sim$400 K a minor impact of older air is found. This is in agreement with the CLaMS backward-trajectory calculations, that results in a BL fraction of a few percent below 100% at these levels of potential temperature (Fig. 4a). Mixing with older stratospheric air above $\sim$400 K is evident in both the trajectory-based mean transport time as well as the mean age of air inferred from 3-dimensional CLaMS simulations.

However, there is a strong difference between the used ECMWF reanalyses as already found in trajectory-based mean transport times at potential temperatures higher than 410 K. ERA-Interim results in a mean age of about 2 years, while using ERA5 $1° \times 1°$ yields a mean age of more than 3 years at 470 K (Fig. 5). The differences from using ERA-Interim and ERA5 $1° \times 1°$ are much larger than from methodological differences to calculate the mean age of air (the clock-tracer mean age, age spectrum–based mean age and spectrum-based mean age including the tail correction; see Sect. 3.3).

Below 400 K potential temperature there is a difference between the trajectory-based mean transport time and the mean age of air of $\approx$80 days and $\approx$130 days using ERA-Interim or ERA5 $1° \times 1°$, respectively. In pure back-trajectory calculations only advective transport is included and mixing is ignored. Moreover, trajectory transit times are truncated to below about 1 year (1 June 2016), which basically excludes influence of downward transport from the stratosphere. However due to the calculation of the mean value of the transport times of many single trajectories, this statistic treatment represents mixing between different air masses. In contrast, in 3-dimensional CLaMS simulations irreversible mixing is included parameterised by the deformation of the large-scale winds (see Sect. 3.3) and, amongst other effects, enhances downward transport from the stratosphere into the troposphere (Konopka et al., 2019). Hence, larger mean age compared to trajectory transit times is to be expected. But also differences in the treatment of the lower model boundary could cause this difference; the age of air tracer is released in the lowest model layer, whereas the trajectory-based mean transport time is related to the top of the model boundary layer



($\zeta$=120 K ∼ 2-3 km above surface following orography). However, the difference between the trajectory-based mean transport time and the mean age of air form 3-dimensional CLaMS simulations below 400 K will not be further discussed here.

## 4.2 Air mass origin and its vertical propagation

For better source attribution of the StratoClim aircraft measurements it is important to identify the source regions at the model BL. During monsoon 2017 most air parcels were released in the northern part of the Indian subcontinent, the Tibetan Plateau, Bay of Bengal and eastern China (Fig. 6), however the details differ between the three data sets. Using ERA-Interim in general more marine sources are found in the western Pacific compared to ERA5 for monsoon 2017. A cluster of air parcels at the model BL is found over the western Pacific caused by typhoon activity at ∼ 20°N 125°E influencing research flight F08 (Fig. A1)

using ERA5 reanalysis (details see (Stroh and StratoClim-Team, 2023)), whereas this typhoon signature is not found using ERA-Interim and is only very weakly represented in ERA5 $1° \times 1°$. Due to the better representation of convection, a slightly higher fraction of air is transported during monsoon 2017 from the model BL using ERA5 (64%) compared to ERA-Interim (63%) and ERA5 $1° \times 1°$ (63%). The frequency distributions for each research flight (F01-F08) for monsoon 2017 using ERA5 are shown in the supplementary Figure A1.

During pre-monsoon 2017 the origins are shifted towards the tropics to the northern Inter-Tropical Convergence Zone (ITCZ) e.g. over the Indian Ocean and the western Pacific (see supplementary Fig. A2). For winter 16/17, the origins move further to the south to the southern Inter-Tropical Convergence Zone (ITCZ) mostly over the Warm Pool region, northern Australia and western Pacific. The contributions from post-monsoon 2016 and monsoon 2016 are minor. Differences between ERA5 and ERA-Interim are obvious in transport time e.g. during pre-monsoon 2017 and winter 16/17, 25% of air is from model BL using

ERA-Interim and only 20% using ERA5. Thus faster ERA-Interim vertical velocities in the UTLS (as already shown in Fig. 4) have an impact on the spatial distribution of the air mass origin in the model BL and yield differences between ERA-Interim and ERA5. During pre-monsoon 2017 the highest frequency distributions using ERA5 are found in the Bay of Bengal, whereas in ERA-Interim high fractions are found in continental Asia, Bay of Bengal and the tropical western Pacific.

Due to the different vertical velocities in ERA-Interim, ERA5 and ERA5 $1° \times 1°$ (Fig. 4), the propagation of air masses from

different model BL regions into the lower stratosphere varies in the region of the Asian monsoon. To infer these differences the regional mask (Fig. 3) introduced in Sect. 3.2 is applied. Fig. 7 shows the fraction of air from the model BL, split into the BL regions (Fig. 3) as well as the fractions of the free atmosphere using three data sets. The fractions of air are accumulated back to starting times of different seasons: monsoon 2017 (a), pre-monsoon 2017 (b), winter 16/17 (c), post-monsoon 2016 (d), monsoon 2016 (e). The longer the trajectories the higher are the contributions from the model BL and the lower are the

fractions from the free atmosphere. The quality of the reconstruction of $CO_2$ depends on the trajectory length, therefore it is important to know the contributions from the model BL in each altitude. The trajectory length can be too short (and thus miss contributions from the model BL) or too long (resulting in higher uncertainties) (for a detailed discussion on this issue see Vogel et al., 2023).





Below 380 K, in general contribution from continental regions (Indian Subcontinent, Bangladesh, Tibetan Plateau and the
continental northern hemisphere) are higher using ERA5 compared to ERA-Interim (Fig. 8). While using ERA-Interim, higher
contributions from the marine northern hemisphere and the Warm Pool region are found in these levels of potential temperature.
Between 380 K and 420 K, the contributions are vice versa and more marine sources (marine Northern Hemisphere, Warm
Pool region) are found using ERA5 than using ERA-Interim. In particular above 430 K the impact of the tropical southern
hemisphere is much stronger in ERA-Interim compared to ERA5 (Fig. 8).

Using ERA5 $1° \times 1°$ data to drive the CLaMS trajectories yields comparable results using ERA5 reanalysis (Figs. 7 and 8)
and differences between ERA5 and ERA5 $1° \times 1°$ are in general much lower than 10 percentage points in contrast to differences
between ERA5 and ERA-Interim ($\sim$20 percentage points). However the vertical dispersion in ERA5 $1° \times 1°$ is higher as in
ERA5 and ERA-Interim (in particular above 440 K) caused by the down-scaling to a $1° \times 1°$ horizontal grid and a 6-hourly
time resolution loosing some details of upward transport along the trajectories.

## 4.3 Effective ascent rates

From CLaMS trajectories effective ascent rates are calculated as difference in potential temperature along the backward tra-
jectories for a time interval of 1 day and 20 days before the time of the aircraft measurements. The effective ascent rate is an
integrated quantity, depends on time and is not an instantaneous ascent rate at a specific location in the atmosphere. The effec-
tive ascent rates are calculated for the three data sets (ERA-Interim, ERA5 and ERA5 $1° \times 1°$; see Fig 9). Negative effective
ascent rates reflect the descent of air masses just before the flight such as descending air from the lower stratosphere mixing
into the air of the Asian monsoon anticyclone.

The effective ascent rates for 1 day reflect the short term evolution of an air mass and can be impacted by recent convec-
tive events (e.g. Fig. 9b at 390 K) or stratospheric intrusions i.e. mixing with older stratospheric air (Fig. 9b at $\sim$415 K and
$\sim$435 K). Therefore, strong differences between the three data sets are found (Fig. 9a-c). Less impact of convection as well as
of stratospheric intrusions are found in ERA-Interim and in ERA5 $1° \times 1°$. Below 360 K effective ascent rates over 1 day in
ERA5 up to $\sim$50 K/d, in ERA5 $1° \times 1°$ up to $\sim$30 K/d and in ERA-Interim only up to $\sim$20 K/d are found (not shown here).

The mean effective ascent rates over 20 days (Fig. 9d-e) reflect a time-averaged ascent rate which is impacted by (vertical as
well as horizontal) mixing of air masses of different origin and age. In general, the mean effective ascent rates inferred from
ERA-Interim are higher compared to ERA5 and ERA5 $1° \times 1°$. To evaluate the mean effective ascent rates calculated from
CLaMS back-trajectories, mean ascent rates from air samples collected with the whole air sampler (WAS) of Utrecht University
are estimated using measurements of the long-lived trace gases HFC-125 and $C_2F_6$. Both trace gases are chemically inert in
the troposphere and stratosphere and have been demonstrated to be suitable to derive observation-based mean age ages of air
as both have very long atmospheric lifetimes (HFC-125 >800 years (Leedham Elvidge et al., 2018) and $C_2F_6 \approx$10.000 years
(Worton et al., 2007)). As the concept of age of air inferred from measurements only works in the stratosphere, a reference
level of 390 K (corresponding to a mean age of 0 as determined via polynomial fit functions, Figure 10) is used. The mean
ascent rate for each air sample is then simply derived from dividing the potential temperature difference to this reference level
by the age of air derived from the two tracers (Figure 10).







**Figure 6.** Frequency distribution (number of trajectories normalised by the total number of trajectories started along the flight path) of the locations where air parcels were traced back to the model BL. Trajectories driven by ERA-Interim, ERA5 and ERA5 $1° \times 1°$ reanalysis were started along the entire flight paths (every 1 second) of all eight Geophysica flights. The frequency distributions are shown for monsoon 2017 (a zoom of Asia marked as grey box is shown right beside). The frequency distribution is calculated in longitude-latitude bins of $2.0° \times 1.5°$. The percentages indicate the fraction of air parcels released at the model BL within monsoon 2017. In summary, 90% (93%) using ERA5 (ERA-Interim) of the air parcels were released at the model BL after 1 June 2016, the other 10% (7%) originates from aged air. The detailed pattern of the frequency distribution depends on the used reanalyses.







**Figure 7.** The fraction of air from the model boundary layer (BL) and the free atmosphere. The fraction from the model BL and from the free atmosphere is calculated from all backward trajectories started along the Geophysica flight tracks averaged in 2 K intervals and accumulated back to the start times of different seasons (rows), namely monsoon 2017, pre-monsoon 2017, winter 16/17, post-monsoon 2016, monsoon 2016 (detailed start times are listed in Tab. 1) and for three data sets (ERA-Interim, ERA5 and ERA5 $1° \times 1°$; columns). The fraction of air for monsoon 2016 (last row) referred to as the free atmosphere corresponds to the fraction of 'aged air' defined in Tab. 1. The fraction of air from the model BL is divided in the different BL regions as shown in Fig. 3.





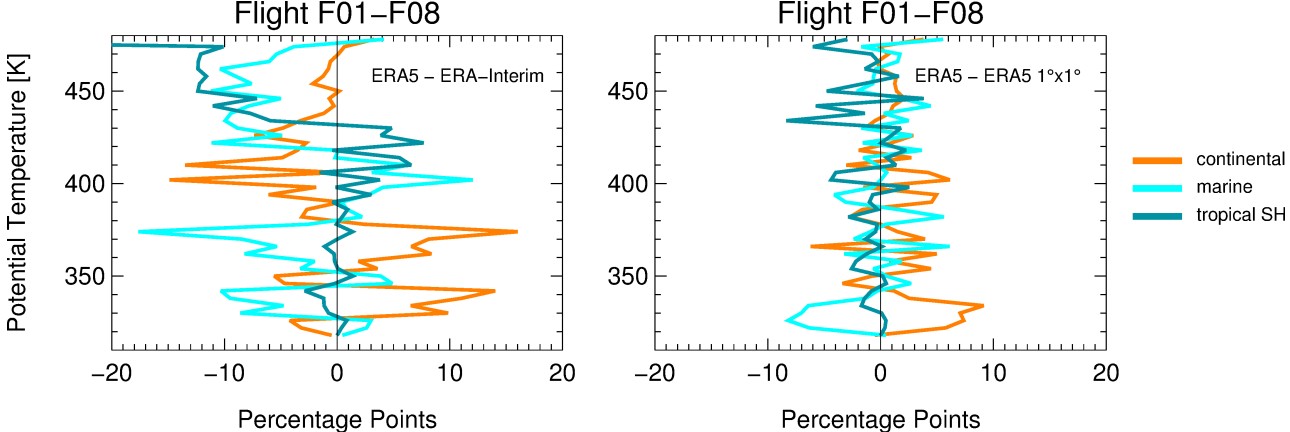

**Figure 8.** The difference of the fraction of air from the model boundary layer (BL) between back-trajectories driven by ERA5 and ERA-Interim (left) as well as between back-trajectories driven by ERA5 and ERA5 $1° \times 1°$(right) depending on potential temperature. The difference is accumulated back to monsoon 2016 (1 June 2016) and fractions are averaged in 4 K intervals in contrast to Fig. 7 (last row; where 2 K intervals are use) to high-light the main impact of the BL sources (however the values of percentage points depends on the used averaging interval). The model boundary layer regions (Fig. 3) are summarised into three regions: continental (India, BGD, TIB, cNH) and marine regions (mNH, Wpool) mainly from the northern Hemisphere as well as the tropical Southern Hemisphere (tSH).

The mean age of air reflects an integrated three-dimensional transit time, impacted by vertical and horizontal transport as well as mixing processes. These processes are likely to be increasingly influential the further away the air is from the 390 K reference surface, with a clear tendency to increase the observation-based mean age of air beause of the horizonal transport (in-mixing) of aged stratospheric air. Therefore, observation-based mean ascent rates are lower limits compared to an idealised air parcel ascending without any mixing with aged stratospheric air. In the trajectory-based mean effective ascent for 20 days vertical and horizontal transport as well as mixing processes are included and therefore comparable with observation-based ascent rates.

Between 390 and 430 K, there is a variability of the mean effective ascent rates derived from HFC-125 and $C_2F_6$ from 0.2 up to 2.3 K/d. Above 430 K, the observation-based ascent rate converges to ~0.2 K/d. Here, the mean effective ascent rate derived from ERA5 (as well as ERA5 $1° \times 1°$) back-trajectories over a time interval of 20 days ($\sim$ 0.2-0.3 K/d) is in good agreement with observation-based mean ascent rates derived from air samples collected by the whole air sampler. Mean effective ascent rates derived from ERA-Interim back-trajectories are much faster ($\approx$ 0.5 K/d) above 430 K.

The observation-based mean age of air at 480 K is about ~2-2.5 years, therefore ERA-Interim mean age is most likely too young (Fig. 5). In contrast using ERA5 $1° \times 1°$ yields a mean age of air about 3 years at this potential temperature level and is likely too old in this altitude range.

Our analysis agrees with previous studies, that found consistently that in general the vertical velocities in ERA-Interim are too fast in the tropics (Dee et al., 2011; Ploeger et al., 2012; Schoeberl et al., 2012). In addition, our findings show that the

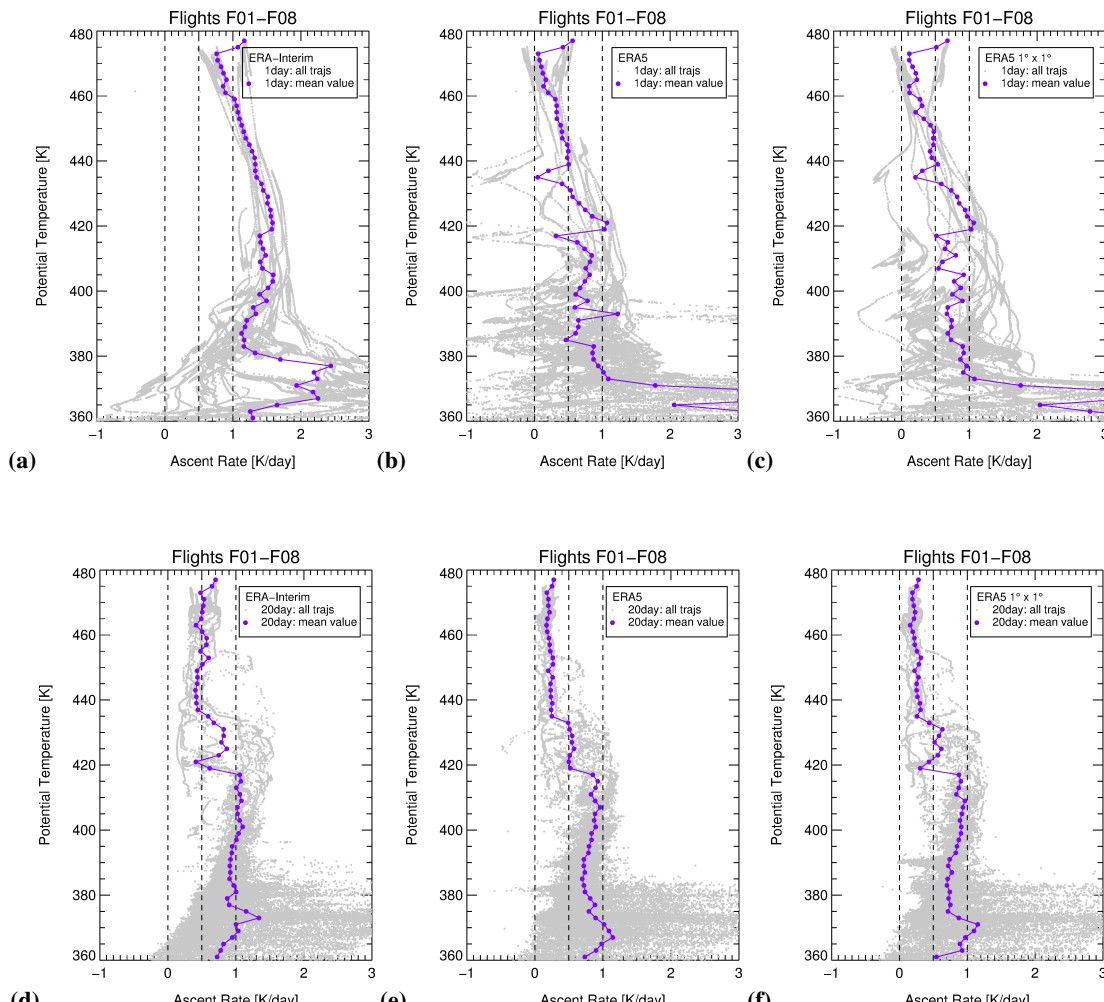

**Figure 9.** Effective ascent rates calculated as difference in potential temperature along backward trajectories driven by three data sets (ERA-Interim, ERA5 and ERA5 $1° \times 1°$) to the time of the StratoClim measurements for a time interval of 1 day (top) and 20 days (bottom) and their mean values in 2 K intervals. The ascent rates are calculated for all trajectories calculated for research flights F01-F08. Negative effective ascent rates reflect the descent of air masses.



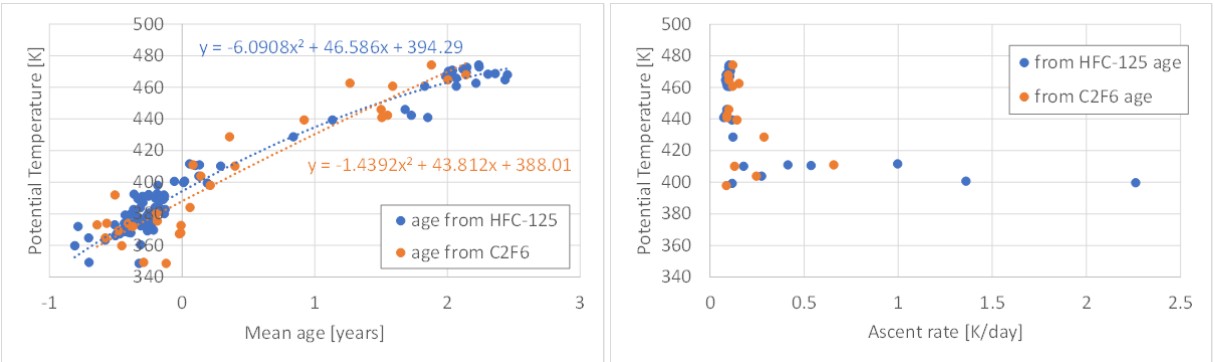

**Figure 10.** Mean age (left) and mean ascent rates above 390 K (right) derived from trace gas measurements of air samples collected with the whole air sampler (WAS) of Utrecht University during the eight StratoClim research flights over the Indian subcontinent in summer 217.

mean effective ascent rates of $\sim$ 0.2-0.3 K/d derived from ERA5 in the region of the Asian monsoon in the lower stratosphere (430 K-480 K) agree very well with observation-based mean ascent rates derived from long-lived trace gases.

## 4.4    Age spectra

We show that the ascent rates along CLaMS backward trajectories depend on the used ECMWF reanalyses. However they further depend on the considered altitude range. Thus, in general ERA-Interim in the UTLS is faster than ERA5. However, due
to a better representation of convection, air masses can be uplifted faster as well as up to higher levels of potential temperature by convection when using the ERA5 reanalysis compared to ERA-Interim. These differences have an impact on the age spectra on different levels of potential temperature using the three data sets.

Age spectra with a time resolution of 5 and 10 days using the three data sets (Fig. 11) reflect the faster vertical velocities found in the UTLS using ERA-Interim, thus the maximum peak of the age spectrum is in general at shorter transport times
compared to ERA5 and ERA5 $1° \times 1°$. However, at lower potential temperatures the impact of convection which is different in the used reanalyses has to be taken into account. Thus a better representation of convection (visible in the two peaks at 380 K for a time resolution of 5 days; black line) and slower vertical velocities found in ERA5 in the UTLS can result in a spectrum peak at similar transit times to the maximum peak as a coarser resolution of convection and faster vertical velocities found in ERA-Interim. Therefore, the maximum peaks for 390 K and 400 K using ERA-Interim are at similar transport times.


## 4.5    Transport times from CO$_2$ measurements

High-resolution $CO_2$ profiles measured in situ (Fig. 12) reflect the seasonal variability of $CO_2$ at ground level (see Fig. 2). $CO_2$ concentrations are relatively independent from diurnal variations in the UTLS, although $CO_2$ has a strong diurnal cycle near



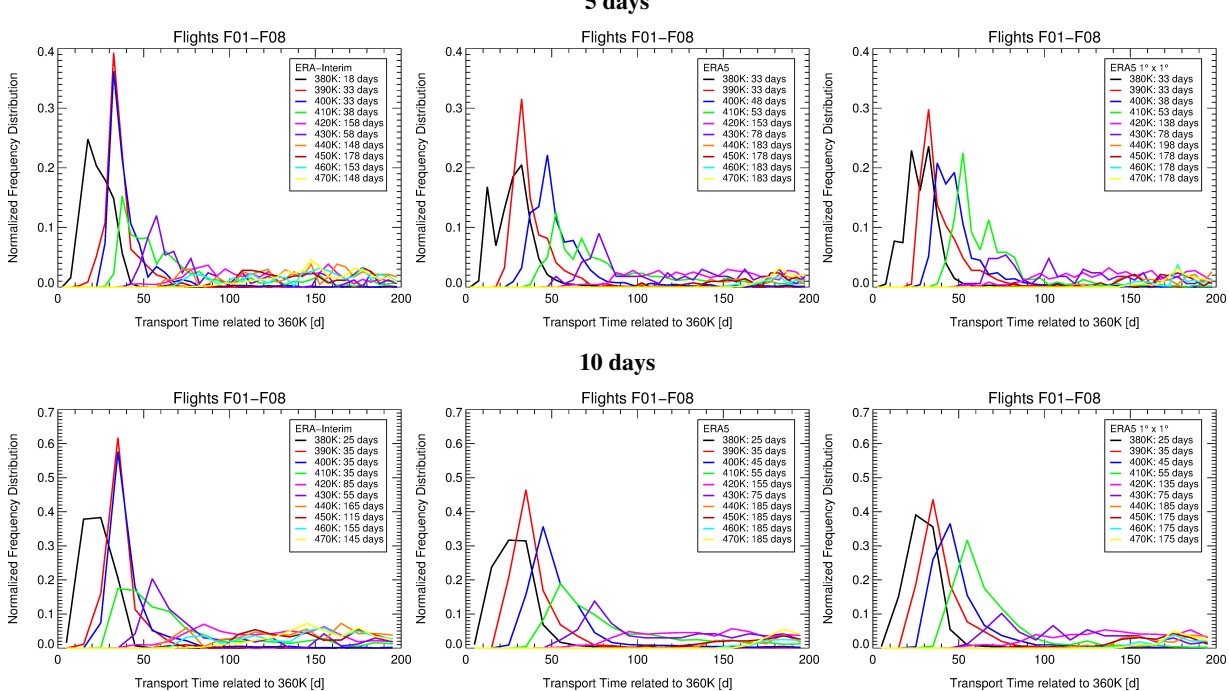

**Figure 11.** Normalised frequency distribution of the transport time from 360 K ($\approx$ the level of maximum convective outflow) to the location of the aircraft measurement along the CLaMS backward trajectories (denoted as age spectrum) using the three data sets. The age spectra are shown for different levels of potential temperature (for 2 K intervals) for a time resolution of 5 and 10 days. In the legend, the transport time to the maximum peak for each level of potential temperature is given.

the ground. Further, $CO_2$ is chemically inert in the troposphere and stratosphere and can be used as an age tracer considering

time periods of several months (e.g. Boering et al., 1996; Andrews et al., 2001; Ray et al., 2022).

     As shown in Fig. 4, trajectory-based transport times increase with the altitude of sampled air parcels. However, there is also a strong variability of transport times between individual air parcels at the same level of potential temperature indicating mixing of air masses of different transport times or of different age (Fig. 12). Moreover, differences in transport times of individual air parcels using the two ECMWF reanalyses are found as well as differences in the tropopause height (in particular for the local

minimum and maximum). Hoffmann and Spang (2022) found that the standard deviation of the tropical tropopause height are $\approx$ 30-50% higher in ERA5 compared to ERA-Interim, related to the higher resolution of ERA5.

     In the following, the trajectory-based transport time is used to infer the mean transport time to the vertical $CO_2$ maximum (Fig. 13). This approach is reasonable because in the Asian monsoon below 410 K mixing with aged air (older than 1 June 2016) is minor (see Fig. 4). The mean transport times (calculated as median in 1 K intervals) using all research flights F01-F08

in the region of the vertical $CO_2$ maximum ($\approx$ 380-400 K) differ using the three data sets in particular at 400 K between 63 d





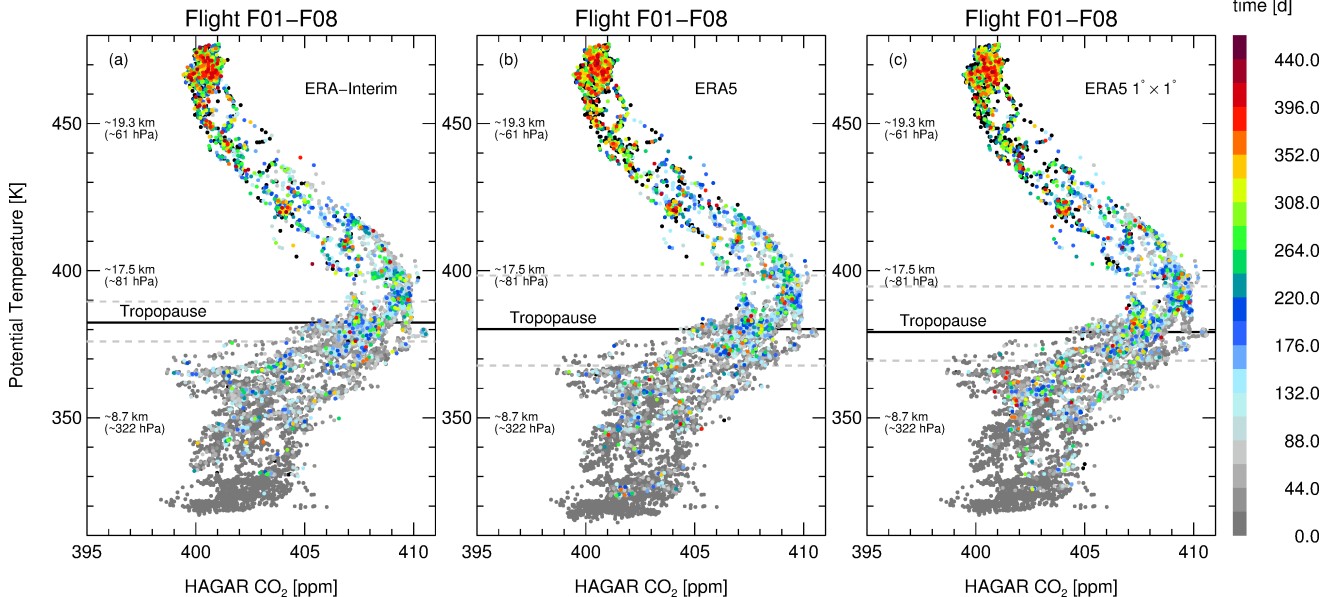

**Figure 12.** Airborne $CO_2$ measurements from the StratoClim campaign in Kathmandu (Nepal) during July and August 2017. Each air parcel is coloured by the trajectory-based transport time from the model boundary layer (BL) to the time of measurements inferred by Lagrangian back-trajectory calculations driven by ERA-Interim (a), ERA5 (b) and ERA5 $1° × 1°$ (c). Air parcels located in the model BL are not shown. Aged air (air located in the free atmosphere on 1 June 2016) is marked in black. In addition, the mean WMO tropopause (Hoffmann and Spang, 2022) as well as the lowest and highest tropopause (grey dashed lines) over Kathmandu during the aircraft campaign (27 July - 10 August 2017) are shown.

(ERA-Interim), 75 d (ERA5) and 78 d (ERA5 $1° × 1°$) as indicated in Fig. 13. A detailed comparison of the trajectory-based mean transport times calculated for each flight is shown in Tab. 2.

The mean transport time ($\overline{T}$) calculated from single flights varies by using the three data sets: 76 d (ERA-Interim), 85 d (ERA5) and 100 d (ERA5 $1° × 1°$), but also here ERA-Interim is fastest. The variability of $\overline{T}$ between the different research
flights F01-F08 is caused by the specific meteorological conditions at the flight day (e.g. high convection during F08 or mixing with older air during F03 and F08) as well as by different flight patterns (more details see supplementary Figure A3).

Using the seasonal cycle of ground-based $CO_2$ measurements in Nainital, a transport time from the $CO_2$ maximum at the ground on 1 May 2017 to the date of the aircraft measurements from 27 July to 10 August 2017, a transport time of 88 to 102 days can be estimated. In addition, the seasonal variability of $CO_2$ over the northern Indian subcontinent (mean value between
20–30°N and 75–95°E) of the lowest model level at 975 hPa of the GOSAT-L4B product (see Sect. 3.2) is used to estimate transport times of the $CO_2$ maximum. The maximum of $CO_2$ in GOSAT-L4B over the Indian subcontinent is around the 15 April 2017, thus a transport time of the maximum of about 103-117 days can be estimated.





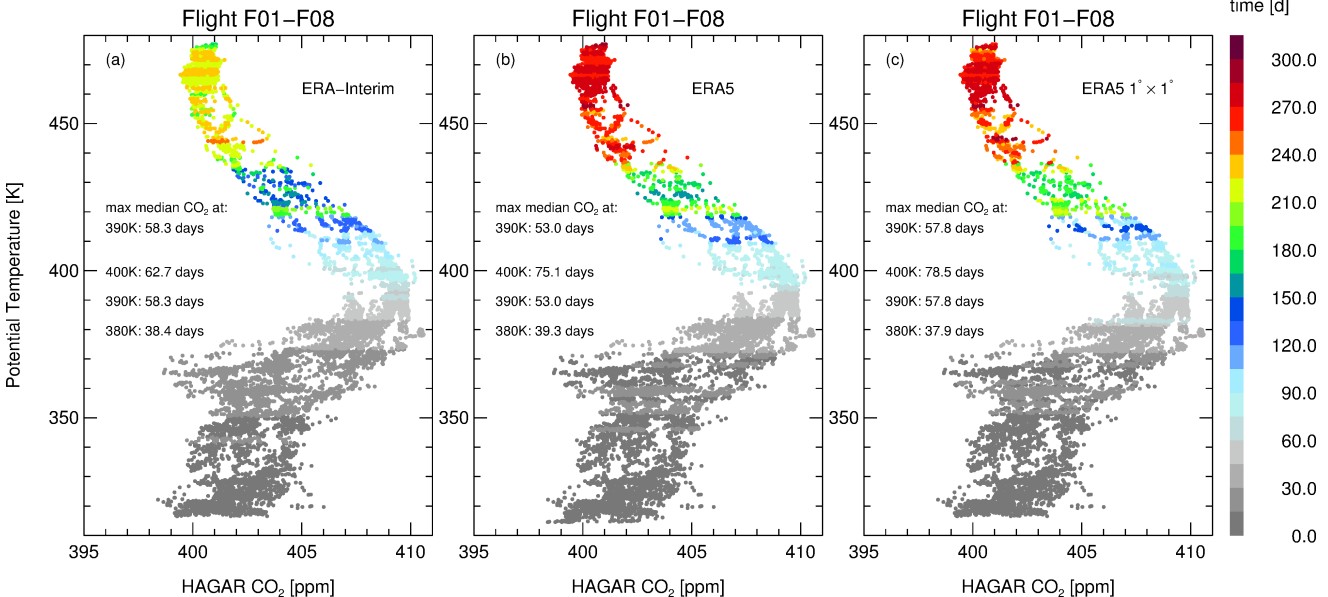

**Figure 13.** Airborne $CO_2$ measurements colour-coded by the mean trajectory-based transport time calculated as median in 1 K intervals of potential temperature. The median is calculated using only trajectories that were released after the 1 July 2016 from the model boundary layer thus excluding aged air which is sufficient up to 410 K. Above 410 K aged air has to be taken into account. The trajectory-based mean transport times at 380 K, 390 K and 400 K are further given. The altitude of the vertical mean $CO_2$ value is here at 390 K potential temperature.

This rough estimate demonstrates that in general the ERA-Interim vertical transport times to the UTLS (here 400 K) are likely too fast compared to the estimate from ground-based $CO_2$ measurements as well as from GOSAT-L4B. However, in this
estimate mixing of air masses from different geographical regions is not taken into account, in particular from Comilla where ground-based $CO_2$ is much more variable as in Nainital (Fig. 2) caused by local land-use (Nomura et al., 2021). To consider also the variability of $CO_2$ at different geographic regions, a detailed $CO_2$ reconstruction is required.

## 4.6 Reconstruction of $CO_2$ from airborne measurements

For a reliable reconstruction of measured vertical $CO_2$ profiles over the entire altitude range, both accurate back-trajectory
calculations are required as well as precise $CO_2$ concentrations at the ground. For the latter purpose, a regional mask was developed where $CO_2$ is prescribed in the model BL depending on different BL regions (Fig. 3).

To reconstruct vertical profiles of trace gases in the region of the Asian monsoon up to 410 K potential temperature the fraction of air from the model BL has to be ~100% otherwise mixing with aged air has to be taken into account. In Sect. 4.1, it was shown that using a trajectory length back to the start time of monsoon 2016 (≈ 10-14 months), a boundary layer fraction

| Flight | date | pot. temperature level | $\bar{t}_{Fi}$ ERA-I | $\bar{t}_{Fi}$ ERA5 | $\bar{t}_{Fi}$ ERA5 $1° \times 1°$ |
|---|---|---|---|---|---|
| F01 | 27.07.2017 | 400 K | 52.8 days | 78.3 days | 73.8 days |
| F02 | 29.07.2017 | 400 K | 57.9 days | 71.6 days | 76.3 days |
| F03 | 31.08.2017 | 400 K | 121 days | 63.6 days | 163 days |
| F04 | 02.08.2017 | 400 K | 93.2 days | 94.7 days | 91.8 days |
| F05 | 04.08.2017 | 400 K | 60.5 days | 76.1 days | 67.9 days |
| F06 | 06.08.2017 | – | – | – | – |
| F07 | 08.08.2017 | 400 K | 69.0 days | 83.7 days | 90.6 days |
| F08 | 10.08.2017 | 400 K | 74.6 days | 129 days | 133 days |
| $\overline{T} = (\sum_{i=1}^{7} \bar{t}_{Fi})/7$ | | 400 K | 75.6 days | 85.3 days | 99.5 days |
| Fig. 13 | | 400 K | 62.7 days | 75.1 days | 78.5 days |

**Table 2.** Trajectory-based mean transport time ($\bar{t}_{Fi}$) to 400 K potential temperature for each research flight calculated as median at 400 K in a 1 K interval. Only air parcels younger than 1 June 2016 are used. Further, the mean transport time ($\overline{T}$) over all flights (($\sum_{i=1}^{7} \bar{t}_{Fi}$)/7) is calculated, whereby flight F06 is excluded because here the maximum flight height is only 380 K. The trajectory-based mean transport time to 400 K potential temperature indicated in Fig. 13 is also listed.

of about 100% is reached up to 410 K. Above 410 K mixing with older air masses successively occurred and the fraction from the model BL rapidly decreases.

    Fig. 14 (top) shows reconstructed $CO_2$ using three data sets for back-trajectory calculations until 1 June 2016 neglecting the contributions from the free atmosphere (aged air). The comparison with measured in situ $CO_2$ profiles shows a good overall agreement from the model BL up to ∼410 K for ERA5 trajectories. A reconstruction using ERA-Interim shows a stronger

dispersion between 390 K and 420 K, because here ERA-Interim vertical velocities are faster than ERA5. Further, below 370 K the measured variability of $CO_2$ caused by convection (low $CO_2$ at ≈ 360 K) is better reproduced using ERA5 due to the better representation of convection. A $CO_2$ reconstruction using trajectories driven by ERA5 $1° \times 1°$ is somewhere between ERA-Interim and ERA5.

    Above ∼410 K, aged air has to be taken into account as discussed in Sect. 4.1. Figure 14 (bottom) shows reconstructed $CO_2$,

but using in addition GOSAT-L4B $CO_2$ data for the fraction of aged air. For back-trajectories ending the free atmosphere $CO_2$ is reconstructed from GOSAT-L4B data that are providing $CO_2$ values up to 10 hPa (for details see Vogel et al., 2023). Here, for each 1 K interval the median of all air parcels considering both the fraction from the model BL as well as from the aged air is calculated. This approach allows the mixing of air at the top of the Asian monsoon anticyclone between air mass from the boundary layer and air from (stratospheric) background to be considered.

Caused by too fast vertical velocity in the UTLS in ERA-Interim (according to effective ascent rates inferred from whole air sampler measurements) higher $CO_2$ from the model BL is found in the lower stratosphere. Even including the contribution of aged air from GOSAT-L4B data yields slightly higher reconstructed $CO_2$ above 420 K using ERA-Interim compared to the measurements. Using ERA5 as well as ERA5 $1° \times 1°$ yield slightly lower reconstructed $CO_2$ above 420 K. However, above 410



K the quality of the GOSAT-L4B needs also to be taken into account for an assessment of the quality of $CO_2$ reconstruction. GOSAT-L4B data depend on $CO_2$ fluxes at the Earth's surface (GOSAT-L4A data), on model resolution as well as on vertical transport in the used atmospheric transport model, which could have a too fast transport in the lower stratosphere, similar as ERA-Interim.

In summary, there are differences in $CO_2$ reconstruction using the three data sets, whereby the statistical variability of the $CO_2$ reconstruction is in the range of the measurements. However, a $CO_2$ reconstruction using ERA5 agrees best with the vertical measured $CO_2$ profile up to 410 K. The UTLS is a very sensitive region regarding the interplay between deep convection and vertical velocities in the lower stratosphere influencing the vertical transport of $CO_2$ and thus the $CO_2$ reconstruction using trajectory calculations.

## 5   Conclusions

It was reported previously that because of a better spatial and temporal resolution, the ERA5 reanalysis yields a better representation of convection than the predecessor ERA-Interim (e.g. Hoffmann et al., 2019; Li et al., 2020; Legras and Bucci, 2020; Malakar et al., 2020). Further it was shown that the vertical transport in ERA-Interim is too fast in the tropical UTLS (Dee et al., 2011; Ploeger et al., 2012; Schoeberl et al., 2012; Tegtmeier and Krüger, 2022). At higher northern-hemispheric stratospheric levels above the tropical tropopause layer, it was reported that ERA5 transport is likely too slow (Ploeger et al., 2021). In general, our findings confirm these results, however in our study, we focus in detail on the Asian summer monsoon region.

Differences in transport of air in the region of the Asian summer monsoon 2017 were inferred using the Chemical Lagrangian Model of the Stratosphere (CLaMS) driven by three data sets, namely two ECMWF reanalyses in different resolution (ERA-Interim, ERA5 and ERA5 $1° \times 1°$). The model results were assessed using unique airborne measurements up to $\sim 20$ km ($\sim 475$ K) during the Asian summer monsoon 2017 conducted with the Geophysica aircraft during the StratoClim campaign in Nepal (Stroh and StratoClim-Team, 2023). CLaMS diabatic backward trajectories were calculated for all Geophysica research flights (F01-F08) performed over the Indian subcontinent. Trajectory-based transport times, origin of air at the Earth's surface, mean effective ascent rates and age spectra as well as mean age of air from 3-dimensional CLaMS simulations were compared using the three data sets.

Below 410 K convection as represented in ERA5 yields faster trajectory-based upward transport (up to $\approx 20$ days) than ERA-Interim. Air masses above 420 K show up to two months ($\approx 60$ days) shorter trajectory-based transport times from the model BL to the UTLS in ERA-Interim compared to ERA5. A better representation of convection and slower vertical velocities above the convection found in ERA5 can yield similar transport times to the UTLS ($\sim 380$ K–390 K) as a coarser resolution of convection and faster vertical velocities are found in ERA-Interim. Therefore, the frequency distribution of the transport time from the level of maximum convective outflow ($\sim 360$ K) to different levels of the aircraft measurement based on back-trajectories (denoted as age spectra) are very sensitive to the three data sets.





**Figure 14.** Reconstructed $CO_2$ using back-trajectory calculations until 1 June 2016 driven by three data sets (ERA-Interim, ERA5 and ERA5 $1° \times 1°$) compared to HAGAR $CO_2$ airborne-measurements. Reconstructed $CO_2$ is shown using the regional mask shown in Fig. 3 for the fraction of trajectories ending in the model BL driven by the three data sets (top). Reconstructed $CO_2$ is shown as median calculated from all trajectories until 1 June 2016 in 1 K intervals for research flights F01-F08. Reconstructed $CO_2$ is shown using in addition GOSAT-L4B $CO_2$ data for the fraction of trajectories ending in the free atmosphere, mainly from stratospheric background (bottom). Bars indicate the range between the 25 and 75 percentile.



Below 380 K, contributions from continental regions (Indian Subcontinent, Bangladesh, Tibetan Plateau and the continental northern hemisphere) to air masses along the flight paths of all 8 local research flights (F01–F08) are higher using ERA5 compared to ERA-Interim while in ERA-Interim higher contributions from the marine northern hemisphere and the Warm
Pool region are found. Although using ERA-Interim for back-trajectory calculations in general more marine sources in these altitudes are identified, the signal from typhoon activity in the western Pacific (in particular during research flight F08 on 10 August 2017) is not visible when using ERA-Interim. Above 380 K, it is the other way round and more marine sources are found using ERA5 and a stronger impact of the tropical southern hemisphere is found using ERA-Interim.

Above 430 K, the mean effective ascent rates derived from ERA5 back-trajectories over a time interval of 20 days ($\approx$ 0.2-0.3
K/day) are in good agreement with the observation-based mean ascent rates inferred from long-lived trace gases such as $C_2F_6$ and HFC-125 derived from air samples collected by the whole air sampler aboard Geophysica. Mean effective ascent rates derived from ERA-Interim back-trajectories are much faster $\approx$ 0.5 K/day at these altitudes. Caused by the difference in the mean effective ascent rates when using two ECMWF reanalyses a different mean age of air is calculated at higher altitudes. At 470 K, a mean age of air of about 2 years is calculated using 3-dimensional CLaMS simulations driven by ERA-Interim,
while using ERA5 $1° \times 1°$ a mean age of more than 3 years is calculated (a 3-dimensional CLaMS simulation driven by ERA5 is not yet available). At these altitudes, observation-based age of air is up to $\sim$2–2.5 years. Therefore the mean age of air using ERA5 $1° \times 1°$ is likely too old in this altitude range, however ERA-Interim mean age is too young. Thus uncertainties regarding the correct simulation of mean age of air in the lower stratosphere (430 K-480 K) still remain.

Trajectory-based transport times inferred from ERA5 $1° \times 1°$ trajectories have in principle a similar behaviour as the times
obtained using ERA5, however the variability is somewhat different, caused by the reduced temporal and horizontal resolution in ERA5 $1° \times 1°$. Details in simulated transport e.g. air mass origin, impact of tropical cyclones, age spectra and the vertical dispersion are different when employing ERA5 and ERA5 $1° \times 1°$. For long-term simulations over several years or decades ERA5 $1° \times 1°$ may be an acceptable (computing-time-saving) data set , however for detailed transport calculations (e.g analysing aircraft or balloon measurements) the full resolution ERA5 reanalysis is recommended to use.

Further, high-resolution $CO_2$ profiles measured aboard Geophysica were reconstructed using ground-based measurements of $CO_2$ mainly from Nainital (northern India) by Lagrangian model simulations using three data sets (ERA-Interim, ERA5 and ERA5 $1° \times 1°$) leading to an improved understanding of the vertical structure of $CO_2$ in the monsoon region. A reliable reconstruction (simulation) of vertical $CO_2$ profiles during the Asian monsoon is a challenge for model simulations because the seasonal variability of $CO_2$ at the ground, mixing with aged stratospheric air as well as the vertical velocities (including
convection as well as vertical ascent caused by diabatic heating in the UTLS) have to be simulated accurately. Our analysis shows that by using the ERA5 reanalysis for $CO_2$ reconstruction a slightly better agreement with high-resolution in situ aircraft $CO_2$ measurements is obtained compared to ERA-Interim. However at higher altitudes (above 410 K), uncertainties remain in the used reconstruction approach, mainly caused by the limitations of the GOSAT-L4B $CO_2$ data used for the characterising aged stratospheric air, demonstrating the need for better global $CO_2$ simulations. Further, a sufficiently dense coverage of
continuous quality-controlled ground-based monitoring of $CO_2$ over the Indian subcontinent is a prerequisite for a reliable simulation of vertical $CO_2$ profiles in the region of the Asian summer monsoon.



*Code and data availability.* The StratoClim data can be downloaded from the HALO database at https://halo-db.pa.op.dlr.de/mission/101. For more details on the measurements please contact C. Michael Volk (M.Volk@uni-wuppertal.de) for HAGAR $CO_2$ and Johannes Laube (j.laube@fz-juelich.de) for $C_2F_6$ and HFC-125 whole air sampler measurements. Ground-based $CO_2$ from Nainital and Comilla were

provided through National Institute for Environmental Research (NIES) available under Terao et al. (2022a, b). Ground-based $CO_2$ measurements from other sites can be downloaded from the World Data Centre for Greenhouse Gases (WDCGG) (https://gaw.kishou.go.jp) and GOSAT-L4B $CO_2$ data under (https://data2.gosat.nies.go.jp/index_en.html). The ERA-Interim and ERA5 tropopause data are available under Hoffmann, Lars; Spang, Reinhold, 2021, "Reanalysis Tropopause Data Repository", https://datapub.fz-juelich.de/slcs/tropopause. The CLaMS trajectory code is available on a GitLab server at https://jugit.fz-juelich.de/clams/CLaMS.



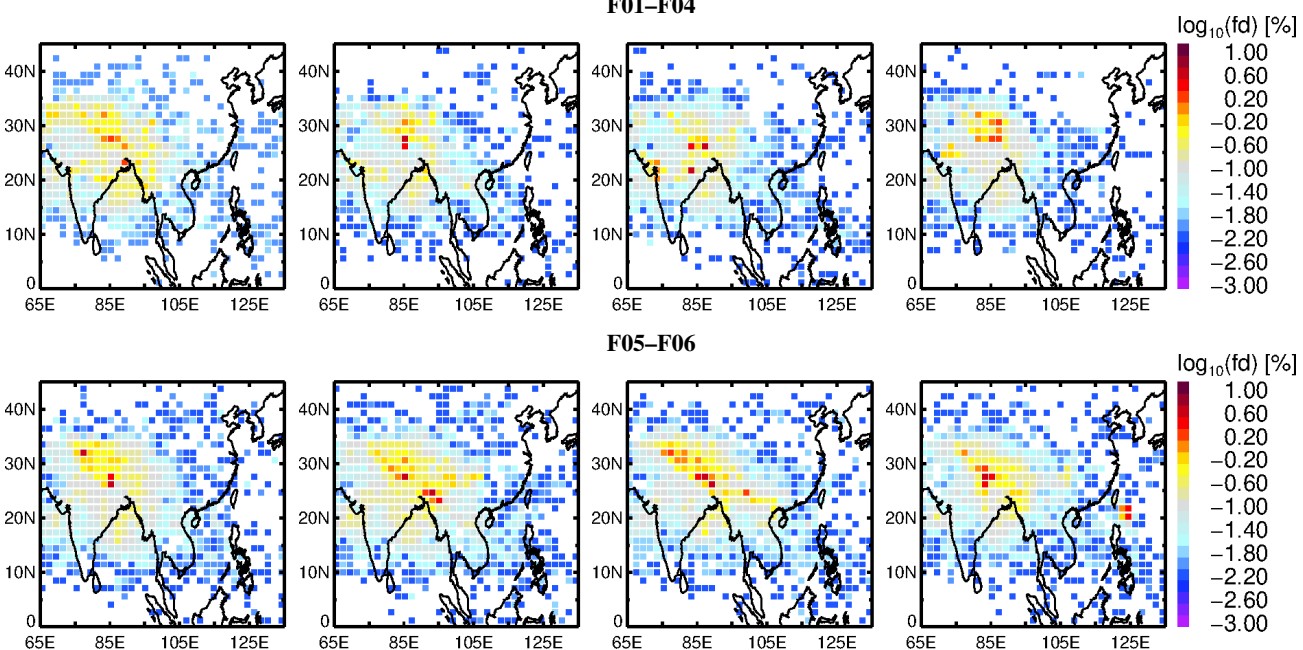

**Figure A1.** Frequency distribution (fd) of the air mass origins at the model boundary layer (BL) for each research flight F01-F08 using ERA5 reanalysis for monsoon season 2017.

**Appendix A:  Supplement: Air mass origin and trajectory-based transport time**

*Author contributions.*  FS was leading the organisation and coordination of the StratoClim aircraft campaign. CMV, JW and VL were responsible for the measurements and analysis of airborne $CO_2$ profiles. JL was responsible for the observation-based mean age as well as ascent rates and FP for age of air from 3-dimensional CLaMS simulations. LH provided tropopause altitudes. FP, GG, JC, JG and LH helped with provisioning of ECMWF reanalyses. CLaMS trajectory calculations and $CO_2$ reconstruction were performed by BV. The study was
conceived by BV, CMV and RM, whereby the results were discussed by all co-authors. The paper was written by BV with contributions from all co-authors.

*Competing interests.*  At least one of the (co-)authors is a member of the editorial board of Atmospheric Chemistry and Physics.

*Acknowledgements.*  The authors are indebted to many local institutions, authorities, as well as individuals for making the StratoClim aircraft field campaign a success. We are especially grateful to the Nepalese, Indian, and Bangladeshi authorities for granting clearances as


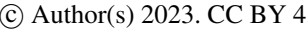


**Figure A2.** Frequency distribution (fd) of the air mass origins at the model boundary layer (BL) similar as Fig. 6, but for pre-monsoon 2017, winter 16/17, post-monsoon 2016 and monsoon 2016. The percentages indicate the fraction of air parcels released at the model BL within a certain season. The detailed patterns of the frequency distribution depend strongly on the considered season as well as on used three data sets (ERA-Interim, ERA5 and ERA5 $1° \times 1°$).

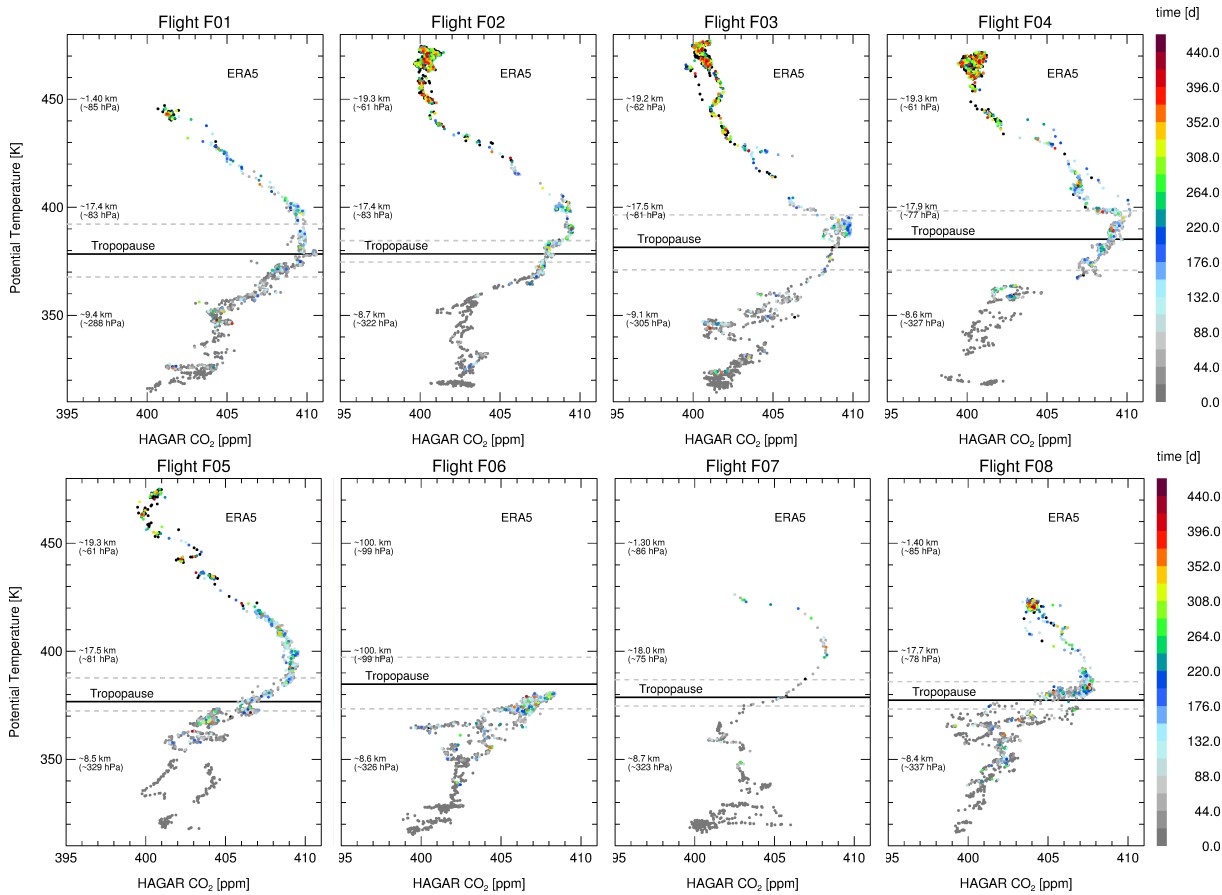

**Figure A3.** Airborne $CO_2$ measurements from the StratoClim campaign in Kathmandu (Nepal) and trajectory-based transport time for each research flight F01–F08 using ERA5 reanalysis (similar as Fig. 12, but for single flights). Each air parcel is coloured by the trajectory-based transport time from the model boundary layer (BL) to the time of measurements. Air parcels located in the model BL are not shown. Aged air (air located in the free atmosphere on 1 June 2016) is marked in black. In addition, the mean WMO tropopause (Hoffmann and Spang, 2022) as well as the lowest and highest tropopause (grey dashed lines) over Kathmandu on the day of the measurement are shown inferred from ERA5.



well as the Kathmandu airport authorities for their local support. Strong support by several local science partners is highly appreciated. We thank the Geophysica aircraft crews and pilots. The European Commission has granted and funded the StratoClim project within Framework Programme 7 under ENV.2013.6.1-2, Grant agreement No. 603557. The HAGAR operations and data analysis was supported by Thorben Beckert from University Wuppertal and were partly funded by the German Helmholtz Association within the Helmholtz-CAS Joint Research Group No. 307. The Nainital and Comilla measurements were performed by Manish Naja from Aryabhatta Research Institute of Observa-

tional Sciences, Md. Kawser Ahmed from University of Dhaka, and Shohei Nomura, Toshinobu Machida, Motoki Sasakawa Hitoshi Mukai from NIES, and supported by the Environment Research and Technology Development Fund (grant nos. JPMEERF20152002, 20182002 and 21S20800) of the Environmental Restoration and Conservation Agency of Japan. Further, the authors gratefully acknowledge the World Data Centre for Greenhouse Gases (WDCGG) for providing $CO_2$ ground-based measurements in particular Yong Zhang from China Meteorological Administration, Beijing, China; Kirk Thoning, Pieter Tans, Ed Dlugokencky and Xin Lan from Earth System Research Laboratory

(NOAA), Boulder, US. Further, we would like to thank the Japan Aerospace Exploration Agency (JAXA), the National Institute for Environmental Studies (NIES), and the Ministry of the Environment (MOE) for providing the GOSAT L4B-data product in particular Shamil Maksyutov, the European Centre for Medium-Range Weather Forecasts (ECMWF) for providing ERA-Interim and ERA5 reanalysis and the Jülich Supercomputing Centre (JSC; Research Centre Jülich, Germany) for computing time on the supercomputer JUWELS (project CLaMS-ESM) and for storage resources on the meteocloud data archive. JCL received funding from the ERC (EXC3ITE-678904-ERC-2015-STG).

Finally, we acknowledge our colleagues from IEK-7 (Research Centre Jülich) Mohamadou Diallo, Paul Konopka, Nicole Spelten and Nicole Thomas for support and discussion.





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
