# Peer review of "Evaluation of vertical transport in ERA5 and ERA-Interim reanalysis using high-altitude aircraft measurements in the Asian summer monsoon 2017"

_EGUsphere, 2023_

## Author Comment (AC1)

**Author Comment to Referee #1**

**Egusphere-2023-1026, 'Evaluation of vertical transport in the Asian monsoon 2017 from CO2 reconstruction in the ERA5 and ERA-Interim reanalysis' by B. Vogel et al.**

We thank Referee #1 for the positive review and for further guidance on how to revise our manuscript. Our reply to the reviewer comments is listed in detail below. Questions and comments of the referee are shown in italics. Passages from the revised version of the manuscript are shown in blue.

*Review of Vogel et al., Evaluation of vertical transport in the Asian monsoon 2017 from CO2 reconstruction in the ERA5 and ERA-Interim reanalysis.*

*The paper by Vogel et al aims at quantifying vertical transport in the UTLS of the monsoon region. They combine in-situ measurements of CO2 with simulations of the Chemical Lagrangian model of the stratosphere (CLaMS) driven by ERA-Interim, ERA5, and 1x1 regridded ERA5 reanalysis data.*

*They apply backward trajectory transport analysis extending backward by more than a year with age of air derived from CLaMS for the different driving reanalysis data sets and compare these with long-lived tracers to infer ascent time scales.*

*They use surface CO2 observations in different regions and combine these with the trajectories and show that the reconstruction using ERA5 gives a good agreement of reconstructed CO2 and measurements up to 410K, Above the reconstruction is affected by mixing with stratospheric air.*

*The authors conclude, that the results are highly sensitive to the representation of vertical transport in the troposphere in the different reanalysis data sets. According to their methods ERA5 yields the most reliable results compared to the observations. Using quasi-inert tracers (C2F6, HFC-125) they their results indicate a good agreement with ascent rates from ERA5 (also 1x1) with large mean age differences at 470 K between ERA-Interim derived age and ERA5 (1x1) of about one year.*

*The paper is well written and the methodology is clearly given. The results regarding the different reanalysis data sets are important for the community, since*

*a lot of conclusions on stratospheric transport were based on ERA-Interim before the release of ERA5. The reconstruction with CO2 is impressive and balanced discussed. Therefore the paper clearly merits publications and I have only a few comments, which are minor.*

We thank Referee #1 for this very positive review. A detailed discussion about the reviewer's minor comments follows below.

**Minor Comments:**

1. *Since a large number of species have been measured at the STRATOCLIM mission, I wondered, if one could include other shorter-lived species to further support the transport time results above the tropopause. In general shorter-lived species should fade out (NMHC) or decrease to background (CO) when being uplifted. I wondered, if the authors thought about including such constituents, which would strengthen their estimates at least above the tropopause.*

   Many thanks for this comment. Yes, it is correct that during StratoClim several shorter-lived species were measured (e.g. Adcock et al., 2021; von Hobe et al., 2021). In our study we focus on trace gases with a very long chemical life time ($CO_2$, HFC-125, $C_2F_6$, $SF_6$) to exclude any chemical effects (e.g chemical reduction) and thus concentrating on transport and mixing. During Stratoclim short-lived species such as dichloromethane, 1,2-dichloroethane and chloroform were measured by the air sampler. The mixing ratios of all three trace gases are decreasing strongly above the tropopause (see Fig. 3 in Adcock et al., 2021). However, for all three substances no published stratospheric life times are available. Further, air samples at the ground and in the troposphere (e.g. Fig. 3 in Adcock et al., 2021) of these trace gases show a very high variability up to tropopause altitudes. This variability make it very difficult to infer transport times (ascent rates) just above the tropopause because of the strong variability of the trace gases around tropopause altitudes.

2. *Fig.2: Could you add the Mauna Loa curve and the classical tropical boundary condition for CO2 at the tropopause as given by e.g. Andrews et*

[Figure]

Figure 1: Temporal variability of ground-based $CO_2$. The variability of ground-based $CO_2$ is shown at Nainital and Comilla (geographical positions see Fig. 3 of Vogel et al. (2023b)). In addition, the seasonal variability of $CO_2$ over the northern Indian subcontinent (mean value between 20–30°N and 75–95°E) of the lowest model level at 975 hPa of the GOSAT-L4B product for comparison to ground-based $CO_2$ measurements is shown. Further, ground-based $CO_2$ measured in Mouna Loa (Hawaii) and in Cape Matatula (Samoa) as well as their average (black dashed-dotted line) as reference for the tropical background are given. The pre-monsoon period (March–May) when a seasonal $CO_2$ maximum is expected is high-lighted (light-grey) as well as the period of the StratoClim aircraft campaign during monsoon 2017 (dark-grey).

*al., 1999, which is the mean of American Samoa surface cycle and Mauna Loa?*

We agree that it is helpful to add the Mauna Loa and the Samoa $CO_2$ surface cycle as well as the mean of both as shown in Fig. 2 in Andrews et al. (1999). The Mauna Loa and the Samoa $CO_2$ surface cycles are already shown and discussed in Fig. 1 in Vogel et al. (2023a). To avoid too much repetition of Vogel et al. (2023a) in Vogel et al. (2023b), we didn't show the Mauna Loa and the Samoa $CO_2$ surface cycle in Vogel et al. (2023b). However, we agree with Reviewer #1 and added this information to the revised

version of the manuscript as shown in Fig. 1 of this reply. Particularly, the mean of Mauna Loa and the Samoa representing the tropical background is an added-value for our paper. We added the following text to Sect. 3.2 of the revised version of the manuscript.

Ground-based $CO_2$ (provided by the World Data Centre for Greenhouse Gases (WDCGG), `https://gaw.kishou.go.jp`) measured in Mauna Loa (Hawaii) and in Cape Matatula (Samoa) (Thoning et al., 2021, `http://doi.org/10.7289/V5X0659V`) as well as their average (black dashed line) are also shown in Fig. 1 (of this reply) as reference for the tropical background (e.g. Boering et al., 1996; Andrews et al., 1999). The comparison of the different seasonal cycles of the ground-based $CO_2$ measurements demonstrates that the seasonal $CO_2$ maximum over the Indian subcontinent during pre-monsoon is much larger than the $CO_2$ maximum of ground-based $CO_2$ of the tropical background.

3. *l.315-335: Ascent rates: Would it be possible to support the ascent rates (20days) with measured vertical gradients of short-lived species, which should show a considerable decrease over 20 days? This would complement the stratospheric analysis based on the very long-lived species presented in Fig.10.*

   See discussion above to point #1.

4. *Was SF6 available for age calculations?*

   $SF_6$ was measured during the StratoClim campaign by the multi-tracer in situ instrument HAGAR operated by the University of Wuppertal (see Details in Sect. 2 in Vogel et al., 2023b) as well as by the whole air sampler (Adcock et al., 2021). In Asia, $SF_6$ has strong sources, therefore it is difficult to use $SF_6$ as a tracer for mean age of air. Nevertheless we show mean age of air deduced from $SF_6$ measured by the whole air sampler in the revised version of the paper (details see below point #6).

5. *l.392: How reliable is the use of just one location at the surface to derive*

*mean transport time? The authors state in l.400 ff that a detailed CO2 reconstruction using comprehensive data is needed, which makes more sense. I'd recommend to skip l.392-397.*

Following the advice of both reviewers, we removed Sect. 4.5 (L382-402 and Fig. 13) in the revised version of the manuscript. Parts of the text included in Sect. 4.5 as well as Fig. 12 are revised and shifted to Sect. 3.2 and 4.6.

6. *Fig. 10 (and general discussion of mean age of air): How well does CLaMS age of air resembles the observational derived age of air (either by the species in Fig 10, or by CO2 itself or eventually SF6 or N2O)?*

We agree that this is an important question. Therefore, we added a more detailed discussion as well as Fig. 2 (of this reply) comparing observation-based age of air derived from $N_2O$ measurements to simulated age of air to Sect. 4.1 in the revised version of the manuscript:

To validate clock-tracer mean age of air as well as trajectory-based transport times from CLAMS we use $N_2O$ measured by the HAGAR instrument during the StratoClim research flights. We compute mean age of air ($\Gamma$) from measured $N_2O$ using $\Gamma - N_2O$ correlations by Andrews et al. (2001) and Engel et al. (2002) based on aircraft and balloon measurements. We use Eq. 3 by Andrews et al. (2001) derived for $N_2O$ mixing ratios of the year 1997:

$$\Gamma = 0.0566 \times (313 - N_2O[1997]) - 0.000195 \times (313 - N_2O[1997])^2. \quad (1)$$

This $\Gamma - N_2O$ correlation is adapted to $N_2O$ mixing ratios (in ppb) for the year 2017 as follows:

$$N_2O[1997] = N_2O[2017] \times (313/335). \quad (2)$$

[revised manuscript text omitted]

soon region.

7. *Fig.7: Looking at Theta $> 430K$: Which role plays transport and mixing from the TTL and tropical lower stratosphere for the calculation of fractions and further below the transport time estimates, also for the age of air and the CO2 reconstruction?*

In the stratosphere at potential temperature levels above 430 K, the fraction of air originating on the Indian subcontinent is low compared to contributions from other regions in the tropics and of aged air (older than 1 June 2016) from the stratosphere (Fig. 7). This has to be considered in both the $CO_2$ reconstruction and the calculation of age of air. At these altitudes it is important to consider 3-dimensional global long-term CLaMS simulations to calculate mean age of air (Sect. 3.3), because trajectory-based transport times inferred in our study do not cover time scales older than 1 June 2016. This issue is discussed in detail in Fig. 5 in Vogel et al. (2023b). Ditto for the $CO_2$ reconstruction at these altitudes it is important to include source

regions from outside the Indian subcontinent as well as aged air from the lower stratosphere using the GOSAT-L4B $CO_2$ product (for more details see Vogel et al., 2023a).

8. *The CO2 cycle at the tropical tropopause is probably similar as at the monsoon tropopause, but how does this affect the reconstructed values and times?*

Air masses in the Asian monsoon anticyclone are strongly separated by a horizontal transport barrier from the background air of the residual tropical tropopause region (e.g. Ploeger et al., 2015; Vogel et al., 2015, 2019). In the Asian monsoon anticyclone very young air from Asia is transported very fast upwards by convection. Therefore during the Asian monsoon season the $CO_2$ cycle at the Asian monsoon tropopause is dominated by Asian emissions (e.g. measured in Nainital and Comilla) and their seasonal cycle. However, in other regions at the tropical tropopause (outside of the monsoon systems) the air is much more a composite from different tropical surface regions in the Inter-Tropical Convergence Zone. However, here the transport times to UTLS altitudes are in general longer than within the Asian monsoon anticyclone. Fig. 1 of this reply, shows the seasonal variability of ground-based $CO_2$ at different sites in the tropics.

References: Andrews et al., Empirical age spectra for the lower tropical stratosphere from in situ observations of CO2: Implications for stratospheric transport, JGR, 1999, doi/epdf/10.1029/1999JD900150

---

## Author Comment (AC2)

**Author Comment to Referee #2**

**Egusphere-2023-1026, 'Evaluation of vertical transport in the Asian monsoon 2017 from CO2 reconstruction in the ERA5 and ERA-Interim reanalysis' by B. Vogel et al.**

We thank Referee #2 for this very detailed review and for further guidance on how to revise and improve our manuscript. Our reply to the reviewer comments is listed in detail below. Questions and comments of the referee are shown in italics. Passages from the revised version of the manuscript are shown in blue.

*The manuscript describes the transport properties in the Asian monsoon region during the StratoClim campaign in July-August 2017 and compares the results derived from ERA5 and ERA-Interim reanalysis with quantities derived from the observations. At first sight, it looks like an extended appendix of a published paper (Vogel et al., 2023, V2023 herafter) and there are indeed a number of common figures and elements of text but it brings also a number of new useful results beyond the reanalysis comparison. However, these new results are not necessarily well discussed or exploited and there are a number of problems that need to be addressed, each one being relatively minor but resulting in a fairly heavy weight when added together.*

Many thanks to Reviewer #2 to point out that Vogel et al. (2023b) looks like an appendix of Vogel et al. (2023a). We are aware that we use the same trajectory analysis as well as the same $CO_2$ reconstruction technique in both publications. In Vogel et al. (2023a), the $CO_2$ reconstruction technique is introduced using only the ERA5 reanalysis. Further, the fact that $CO_2$ is sparsely monitored in the Asian monsoon region is highlighted.

In contrast in Vogel et al. (2023b), the focus is on the robustness of the representation of transport processes in different reanalyses, with a particular focus on differences between ERA5, ERA-Interim and ERA51$° \times 1°$ data sets and their consequences on trajectory-based transport times and ascent rates. This transport assessment includes mean age of air from global 3-dimensional CLaMS simulations as well as different trajectory-based transport times and associated ascent rates compared with observation-based age of air and ascent rates from long-lived trace gases that go far beyond the study of Vogel et al. (2023a). We revised the manuscript (Vogel et al., 2023b) following the reviewer's advice to make the message of our new results clearer and more consolidated and clarified all misleading

issues.

*General remarks: First, the title does not reflect the content of the paper. One expect a focus on the CO2 reconstruction while it is only used in section 4.6, that is basically one page, among 15 pages of results which are mostly about proportion of boundary layer air in the campaign samples and its age. Besides this, this section is the least conclusive of the results. Therefore the title should be changed to indicate the real focus of the paper. Then, more comparison should be made with previously published results, in particular with Bucci et al. (2000) (B2020 hereafter) who produced results and figures directly comparable to several ones of the manuscript. Third a careful rewriting should be made as a number of sentences are too long, with several subordinates and hard to read, and a number of needed clarifications must be added.*

We thank Referee #2 for her/his helpful comments and would like to appreciate her/his work reading our manuscript carefully. A detailed discussion about the reviewer's detailed comments follows below. We agree that the title is misleading and revised the title as follows: 'Evaluation of vertical transport in ERA5 and ERA-Interim reanalysis using high-altitude aircraft measurements in the Asian summer monsoon 2017'. Further, a more detailed comparison to Bucci et al. (2020) is included (details see below).

**Detailed Comments:**

1. *3.1 The number of backward trajectories used in this study (11000) is small by present standards and certainly limits the statistics that can be produced. For comparison, B2020 used 1000 more trajectories in their study of the same campaign for a better accound of mixing.*

   We agree that the number of TRACZILLA trajectories used in Bucci et al. (2020) is much larger than used in our studies (Vogel et al., 2023a,b). In Bucci et al. (2020) 1000-back-trajectory are released per second using sub-grid scale diffusion. In our approach, we decided to use pure back-trajectory calculation without any additional parametrisation such as sub-grid scale diffusion used in TRACZILLA or small-scale mixing used in 3-dimensional CLaMS simulations. Our approach makes it clearer to analyse the differences between the different reanalysis data sets because no additional parametrisations are used. A detailed comparison between pure CLaMS back-trajectory calculations and using in addition sub-grid scale diffusion can be found in Clemens et al. (2023b,a)

A detailed comparison between the trajectory calculations used in Bucci et al. (2020) and Vogel et al. (2023b) is among others part of the Strato-Clim overview paper (Stroh and StratoClim-Team, 2023). Because Stroh and StratoClim-Team (2023) is not yet available for the public, parts of the TRACZILLA and CLaMS intercomparison are stated here in this reply:

'In general, the results of TRACZILLA and CLaMS trajectory calculations show a good overall agreement in identifying locations for convection of air masses contribution to the StratoClim aircraft measurements in Nepal 2017 although somewhat different quantities are compared and likewise the model set-ups are very different. A more detailed view, however, reveals that the calculations differ in some regions. Here, it needs to be kept in mind that TRACZILLA calculations include more trajectories as well as vertical diffusion. Further, the calculation of the vertical velocities is slightly different. CLaMS use the total diabatic heating rate from the reanalysis forecast, whereas TRACZILLA uses only the radiative contribution (Bucci et al., 2020). As TRACZILLA is used solely for studying transport above the cloud tops where radiation dominates the heating rate, respective differences are expected to be small. The good agreement between the patterns of locations for convection simulated by TRACZILLA and CLaMS is remarkable because of the different model set-ups. TRACZILLA uses diabatic ERA5-based vertical velocities above convective regions, i.e. above cloud top altitudes from geostationary satellites, whereas CLaMS uses ERA5-based vertical velocities for transport from the model boundary layer upwards.'(Stroh and StratoClim-Team, 2023).

2. *3.1 The choice of a transition at p/psurf = 0.3 does not mean 300 hpa over the Tibetan plateau but rather 170 hPa on the average, that is just in the middle of the TTL. This choice is rather infortunate and might have som impact on the results.*

Many thanks for this comment. We are aware that the choice of p/psurf =

0.3 corresponds to a much lower pressure than 300 hPa over high mountain regions. The hybrid vertical coordinate ($\zeta$) in the CLaMS model is defined exactly as proposed by Mahowald et al. (2002) with this particular reference pressure value, ensuring a purely diabatic calculation indeed only above 300 hPa. However, also in the first layers below the vertical coordinate is still close to diabatic, as the transition from $\theta$ to $\sigma$ occurs rather slow with a sin-function (Pommrich et al., 2014, e.g.). Due to the transition into an orography-following $\sigma$-coordinate below 300 hPa, the vertical velocity includes information on convective transport as resolved in the reanalysis vertical wind and total diabatic heating rate (see Pommrich et al., 2014). We clarified the related text in Sect. 3.1 as follows.

However, also in the first layers below the reference level the vertical coordinate is still close to diabatic, as the transition from potential temperature to orography-following vertical coordinate occurs rather slow (e.g. Pommrich et al., 2014). Therefore, the vertical velocity includes information on convective transport as resolved in the reanalysis vertical wind and total diabatic heating rate (see Pommrich et al., 2014).

3. *3.1 In any case, as the backward trajectories are run to the boundary layer, a significant part of the path is under the region of the potential temperature levels and the applied method should be described. In particular, it is very useful to know whether convection is represented and how, and whether this differs along the reanalysis.*

To clarify this point, we added the following paragraph to Sect. 3.1 in the revised version of our manuscript.

The upward transport and convection in CLaMS (in both trajectory calculations as well as in three-dimensional simulations) depends on the employed reanalysis data (ERA-Interim, ERA5, ERA5 $1° \times 1°$) which differ strongly in the representation of convection (e.g. Hoffmann et al., 2019; Li et al., 2020; Clemens et al., 2023b). The differences between ERA5 and ERA-Interim is attributed among other things to the better spatial and temporal resolution of the ERA5 reanalysis, which allows for a better representation of convective updrafts. Therefore, in ERA-Interim convection over Asia is

underestimated compared to ERA5. In our study no additional parametrisation for convection is used for the CLaMS simulations, only the convection already included in the reanalysis is considered.

4. *3.1 It would be useful to know whether the $1° \times 1°$ version of ERA5 is obtained by subsampling or filtering the high resolution version. The fact that the tropopause range (fig. 12) is about the same in the two version suggests the first choice which is not the good one.*

We revised the paragraph (L144-L147) as follows for better clarity. A comment to the used tropopauses can be found below #31.

'In addition, we use ERA5 data in lower resolution referred to as 'ERA5 $1° \times 1°$' (similar to Ploeger et al., 2021; Konopka et al., 2022). For ERA5 $1° \times 1°$, ERA5 data are truncated to a $1° \times 1°$ horizontal grid and a 6-hourly time resolution (same as ERA-Interim). The vertical resolution is the same as in the original ERA5 reanalysis.'

$\longrightarrow$

Further, we use a version of ERA5 with lower resolution referred to as 'ERA5 $1° \times 1°$' (similar to Ploeger et al., 2021; Konopka et al., 2022; Clemens et al., 2023b). ERA5 $1° \times 1°$ data are directly provided by the ECMWF on a $1° \times 1°$ horizontal grid after down-scaling the original data by truncation of the spherical harmonics representation to a $1° \times 1°$ horizontal grid. In addition, the time resolution is down-sampled to every 6 hours, for better comparability with ERA-Interim. However, the vertical resolution is not changed and is the same as in the original ERA5 reanalysis. ERA5 $1° \times 1°$ data are a computing-time-saving alternative to the full resolution ERA5 data and are particularly suited for 3-dimensional global multi-annual CLaMS simulations.

5. *3.2, l 167-168 : It is hard to understand what kind of interpolation can be performed if data are available only at 3 locations over the Asian continent.*

*It is also unclear what is actually used in the reconstructions shown in 4.6.*

We understand that it is maybe difficult to understand all the details of the $CO_2$ reconstruction, that is described in detail in Vogel et al. (2023a). However, we would like to avoid to much repetition from Vogel et al. (2023a) in Vogel et al. (2023b). We agree that three $CO_2$ ground-based measurement sites in Asia is a low number, however more observations are not available in Asia from 2016 to 2017. Therefore we pointed out in Vogel et al. (2023a) that there is a need of more $CO_2$ ground-based measurement sites in Asia.

We revised the text (L167-168) as follows for clarification.

'For that purpose different ground-based observations available on different time scales (monthly, weekly or daily) were interpolated in time on a daily grid.'

$\longrightarrow$

For that purpose different $CO_2$ ground-based observations (all shown in Fig. 3) available on different time scales (monthly, weekly or daily) were interpolated in time on a common daily grid to get a $CO_2$ mixing ratio for every day from each used measurement site for the $CO_2$ reconstruction.

6. *3.2 The choice of the boxes shown in fig.3 is a bit hard to understand regarding the continental boundaries. Why is the box surrounding CL and called Bangladesh extended only to the east to cover Birmania ? Why is the Tibetan Plateau truncated on the north and west. Why is NTL influenced by the Gange and Indus valley agriculture and industry representative of the Tibetan plateau where there is no industry and a delayed vegetation cycle? As far as I can see from V2023, the reconstruction up to 400K depends mostly to the ground seasonal cycle at NTL and the reconstruction performed above might result from a clever design of the region boundaries. It would be useful to know whether this design was done a priori or a posteriori.*

We added the following paragraph to the revised version of the paper for better clarity.

The definition of the different model boundary layer regions are adjusted according to the available measurement sites. Case studies with different regional masks defining the model boundary layer regions were performed and the regional mask was developed according to the best agreement of reconstructed and measured vertical $CO_2$ profiles. Further, the local air mass transport influencing Nainital is taken into account as explained in Vogel et al. (2023a).

The relevant parts of Vogel et al. (2023a) are repeated here: 'Air masses over the Indian subcontinent were transported from the Indian Ocean region during summer (monsoon season) and from the inland during winter, therefore observations in Nainital are strongly affected by anthropogenic emissions from the Indo-Gangetic Plain during summer. Anthropogenic emissions, e.g., of $CO_2$, in the Indo-Gangetic Plain are higher compared to other regions in India (Nomura et al., 2021) caused by the dense concentration of industries (e.g., thermal power plants, steel plants, refineries) as well as by the very high population density in this area (Fadnavis et al., 2016). Thus air masses transported long-range from the south to Nainital, can uptake these emission while passing over the Indo-Gangetic Plain (Nomura et al., 2021).'

7. *3.3 This description again misses to mention what is done in the tropospheric levels below the 0.3 p/psurf transition and whether convection is parameterized and in the same way as the backward trajectories. It misses overall to mention what is calculated. If the ages are sampled along the StratoClim flight tracks in the 3d ClaMS records, the comparisons made in this paper are sound. If the ages are sampled on each level over a wide area and several decades, the basis of the comparison is much more fragile and their meaning is questionable.*

   *7bis) 3.3 198 : This sentence seems copied and pasted from a description related to BDC calculations. The transit time is here from the BL This sentence also does not suggest that the age is sampled on the StratoClim flight tracks.*

*7ter) 3.3 It should be mentioned whether the calculations used here are a by-product of the calculations used in Ploeger et al. (2021) or are new and specific.*

Many thanks for these comments. We agree that this issue is vaguely explained. Therefore we added the following explanation to the revised version of the manuscript.

Similar as for the CLaMS trajectory calculations, convection resolved in the reanalysis vertical winds and total diabatic heating rates are used for the 3-dimensional CLaMS simulation (see Sect. 3.1). Apart from small-scale mixing, the vertical transport in CLaMS trajectory calculations and in 3-dimensional CLaMS simulations is treated in the same way.

Further, the sentence (L198) is revised for clarification.

'Global 3-dimensional CLaMS simulations are used to calculate the age of air spectrum, the distribution of transit times through the stratosphere, at each location in the stratosphere based on chemically inert pulse tracers (e.g. Ploeger et al., 2021). The 60 different tracer pulses...'

$\longrightarrow$

Ploeger et al. (2021) performed global 3-dimensional CLaMS simulations to calculate the age of air spectrum, the distribution of transit times through the stratosphere, at each location in the stratosphere based on chemically inert pulse tracers. In our study, the globally calculated mean age of air by Ploeger et al. (2021) is interpolated along all Geophysica flights paths (F01-F08). Thus a direct comparison to the trajectory-based transport time is possible. In Ploeger et al. (2021), 60 different tracer pulses ...

8. *4. 214 : This sentence is an example tht requires rewritting.*

We revised this sentence (L214) as follows:

'CLaMS diabatic backward trajectories driven by three data sets (ERA-Interim, ERA5 and ERA5 $1° \times 1°$) were started along the entire flight paths (every 1 second) of all Geophysica flights (F01-F08) performed over the Indian subcontinent to infer a trajectory-based transport time from the location of the measurement back to the time when the air parcel was released at the model boundary layer (BL).'

$\longrightarrow$

CLaMS diabatic backward trajectories driven by three data sets (ERA-Interim, ERA5 and ERA5 $1° \times 1°$) were started along all Geophysica flights (F01-F08) to infer a trajectory-based transport time from the location of the measurement back to the time when the back-trajectory reached the model BL.

9. *4. The word "released" for parcels reaching the BL backward in time is somewhat confusing. At least a sentence should be added to define exactly what is meant.*

We revised the sentence (L218-220) as follows:

However, most air parcels were released at the model BL much later than 1 June 2017, e.g. 64% (63%) of all air parcels are from the monsoon season 2017 using ERA5 (ERA-Interim) reanalysis.

$\longrightarrow$

However, most back-trajectories reach the model BL much later than 1 June 2017, which implies that air parcels probed during the Geophysica flights were released at the model BL much later than 1 June 2017, e.g. 64% (63%) of all air parcels are from the monsoon season 2017 using ERA5 (ERA-Interim) reanalysis.

10. *4.1 Fig 4a could be compared to Fig.10 of B2020*

Many thanks for this hint. We added a discussion regarding to Fig. 10 of Bucci et al. (2020) to Sect. 4.2 of Vogel et al. (2023b) (details see below #23.

11. *4.1 The transition at about 370 K for young trajectories is similar to the crossover defined from forward trajectories by Legras et al. (2000). This is not by chance as a discontinuity should also appear in the backward trajectories that represent convection.*

We assume that this comment is related to the publication from 2020 (Legras and Bucci, 2020). We add the following text to L231 of Vogel et al. (2023b).

The transition at $\sim 370\,\text{K}$ corresponds to the crossover level near $364\,\text{K}$ found in the Asian summer monsoon 2017 by Legras and Bucci (2020). The crossover level marks the separation of descending and ascending motion and thus confirms that convection as represented in the reanalysis data is included in CLaMS backward trajectories.

12. *4.1 The age of air is bounded above 400 K for the young trajectories and above 440 K for the old trajectories. This is an effect of the truncature of the age spectrum. In principle, since the age spectrum is calculated and shown in fig.11, it should be possible to perform a tail correction like in 3D ClaMS. Perhaps this is difficult due to the small number of trajectories. As the median is only 250 days for long trajectories extended over more than a year, it means a fairly flat tail in the age spectrum.*

Thanks for this remark! Indeed, the trajectory-based transport time can only include transport times up to the trajectory length, and also only for those trajectories ending in the model BL. Therefore the "age" presented here should be seen more as a transport time of short transport pathways to the main convective outflow and not of a complete age of stratospheric air. Furthermore, the exponential tail correction for the age spectrum has been shown to be a meaningful approximation only for age spectra in the stratosphere, hitherto (e.g. Li et al., 2012; Diallo et al., 2012; Ploeger and

Birner, 2016). If a similar correction can be used for air masses sampled close to convective outflow around the tropopause still needs to be shown. Therefore, we refrain here from applying an exponential tail correction to the transport time distributions. We are also more careful with the usage of the term 'age spectrum' and replaced it by 'transport time distribution' explained in the revised manuscript as follows:

The frequency distribution of the transport time of backward trajectories from the main convective outflow to the sample region is in the following referred to as 'transport time distribution'.

13. *4.1 By the way, why using here a median age and not a mean like in 3D ClaMS ? Does it differ strongly from the mean ?*

Many thanks for this comment. It seems that there is a misunderstanding. In Fig. 5 of Vogel et al. (2023b), the median in 2 K intervals is used for both trajectory based transport times as well as the mean age from global 3-dimensional CLaMS simulations. We revised L244 of Vogel et al. (2023b) as follows to clarify the misunderstanding :

'Considering in addition aged air (older than 1 June 2016) the mean age of air from a 3-dimensional CLaMS simulation is compared to the trajectory-based mean transport times calculated from pure back-trajectory calculations (Fig. 5).'

$\longrightarrow$

Considering in addition aged air (older than 1 June 2016) the mean age of air from a 3-dimensional CLaMS simulation interpolated along the flight tracks (and averaged as median in 2 K intervals) is compared to the trajectory-based mean transport times calculated from pure back-trajectory calculations (Fig. 5).

In this study, we use the median instead of the mean, because the variability on a certain level of potential temperature depends on the flight tracks of the

research flights and reflect non-normal distribution at these altitudes. Therefore, some outliers (e.g. caused by high convection or mixing of different air masses such as stratospheric intrusions) are found on a fixed potential temperature, which are better statistically treated using the median instead of the mean (see Fig. 12 and A3 in Vogel et al. (2023b)).

14. *4.1 It is not easy to understand exactly what is shown by the gray area. Are the boundaries of this area the two alternate age curves ?*

Yes, it is. We revised the caption of Fig. 5 as follows:

'Methodological differences in calculating mean age of air are indicated as shading, showing in addition the difference between the age spectrum–based mean age and the spectrum-based mean age including the tail correction (Sect. 3.3).'

$\longrightarrow$

Methodological differences in calculating mean age of air are indicated as shading (in light-blue and light-grey). The left envelope represents the age spectrum–based mean age and the right envelope the spectrum-based mean age including the tail correction (Sect. 3.3).

15. *4.1 An interesting result of Fig. 4a is that long trajectories are needed to reproduce a pattern of the BL fraction that matches the N2O curve shown in V2023. This has some implication for the confinement of the Asian Monsoon Anticyclone (AMA). As there is almost no in-mixing up to 400-420 K (Vogel et al., 2019, Legras et al., 2020). This means that the old tropospheric N2O air was captured when it was formed in 2017 and kept inside.*

Many thanks for this comment. We agree, that this is an interesting result. According to the $N_2O$ vertical profile measured by the HAGAR instrument during StratoClim (shown in Fig. 6 in the revised version of our paper) and our CLaMS trajectory calculations, at least air masses from pre-monsoon 2017 and winter 16/17 have to be taken into account to characterise the air masses between 400 and 420 K at the top of the Asian monsoon anticyclone

(see Fig. 7 in Vogel et al. (2023b)). Therefore, at these altitudes it is important for the $CO_2$ reconstruction to include also ground-based measurements outside from Asia, i.e. from other tropical regions, however the detailed transport times depend on the used reanalysis (see Sect. 4.2 and 4.6).

16. *4.1 The drop of the BL fraction from about 90 % at 370 K to about 0 % at 420K in two months for young trajectories is compatible with a mean ascent rate of about 50/60 = 0.8 K/day inside the AMA. This is smaller that the estimate of Legras et al. (2020) which is 1.1 K/day but larger than the estimate made in 4.5.*

Many thanks for this comment. It is not possible to compare mean ascent rates derived by Legras and Bucci (2020) and Vogel et al. (2023b) directly, because we focus here on backward-trajectories started along the Geophysica flight tracks in contrast to Legras and Bucci (2020) who started a bunch of forward trajectories in the entire Asian monsoon region. However, the relative differences between ERA5 and ERA-Interim are consistent between Legras and Bucci (2020) and Vogel et al. (2023b); in both studies ascent rates derived from ERA-Interim are faster than from ERA5 in the lower stratosphere.

In Fig. 4 in Vogel et al. (2023b) young air masses with an age lower than 2 months are shown. A calculation of a mean ascent rate of 0.8 K/d is not the correct mean ascent rate because also older descending air masses have to be taken into account at these altitudes as shown in Sect. 4.3 (Fig. 9) in Vogel et al. (2023b). According to our understanding that would be in agreement with Legras and Bucci (2020) demonstrating that above a crossover level near 364 K descending air has to be taken into account. Therefore, in Sect. 4.3 we introduce effective ascent rates depending on a certain time interval as well as on a potential temperature level. In Fig. 9 in Vogel et al. (2023b), an effective ascent rate between 370 K and 420 K of about 1 K/d (ERA-Interim) and 0.7-1.0 K/d (ERA5 and ERA5 $1° \times 1°$) was calculated averaged over the last 20 days including also descending air masses. According to our understanding, in Legras and Bucci (2020) the age of an air parcel is defined as the duration elapsed from its release by convection. Figure 5 in Legras and Bucci (2020) shows how the convective impact propagates inside the Asian monsoon anticyclone from the sources as a function

of age and a mean ascent rate of the convective signal of 1.1 K/d was calculated using ERA5. It seems that mixing with older air masses from outside the Asian monsoon is not included in this approach. Because we focus here on backward-trajectories started along the Geophysica flight tracks, the sampling of the air masses during the flights include i.a. stratospheric intrusions, that may have an impact on the effective ascent rates presented in Vogel et al. (2023b). Nevertheless, a mean ascent rate of 1.1 K/d inferred in Legras and Bucci (2020) within the Asian monsoon anticyclone is not in direct contradiction to an effective ascent rate of 0.7-1.0 K/d averaged over 20 days between 370 K and 420 K inferred for air masses sampled during the StratoClim research flights.

17. *4.1 l.250 : Specify your are commenting the old trajectories curve*

In Sect. 4.1 (L250) is stated:

'$N_2O$ profiles measured during the StratoClim campaign indicate strong mixing with older stratospheric air above $\sim 400$ K (Vogel et al., 2023a). '

It is unclear to the authors what the comment 'the old trajectories curve' is related to. Maybe there is a wrong line number?

18. *4.1 l.260 : The sentence starting with "In pure back-trajectory ..." is distracting and could be removed*

In the revised version of the manuscript, the sentence (L262) is removed:

'In pure back-trajectory calculations only advective transport is included and mixing is ignored.'

19. *4.1 l.264 : As far as turbulent diffusive mixing is concerned, that is the type of mixing represented by the basic algorithm of ClaMS, it should not influence the dispersion of a group of parcel beyond a few days. After this diffusive time, the dispersion is due to shear and strain and becomes exponential in time. If now the convective mixing parameterized in Konopka et*

*al. (2019) is concerned, longer time effects are expected. This is were an accurate description of what is done regarding convective mixing in section 3 is missing.*

We agree that a better description about convection and mixing is useful. Therefore Sect. 3 is revised according to the reviewer's advice (see author's comment to #3 and #7). Further the following part (L262-268) is revised for clarification.

'In contrast, in 3-dimensional CLaMS simulations irreversible mixing is included parameterised by the deformation of the large-scale winds (see Sect. 3.3) and, amongst other effects, enhances downward transport from the stratosphere into the troposphere (Konopka et al., 2019). Hence, larger mean age compared to trajectory transit times is to be expected.

$\longrightarrow$

The 3-dimensional CLaMS simulations (see Sect. 3.3) used here to calculate the mean age of air also include parameterised small-scale mixing (dependent on the deformation rate in the large-scale flow) which causes an additional ageing of air compared to the pure trajectory calculations (e.g. Konopka et al., 2019).

20. *4.1 The final discussion and final sentence of this section are somewhat delusive. Why showing the curves from 3D ClaMS in Fig.5 if no conclusion is drawn besides some technical comments ? If the authors believe that the differences could be explained on such technical basis, this should be thoroughly tested as it jeopardizes all the discussion. I believe they are not but more discussion is needed here.*

Many thanks for this comment. We agree and removed the following sentence (last sentence of Sect. 4.1) in the revised version of the paper:

'However, the difference between the trajectory-based mean transport time and the mean age of air form 3-dimensional CLaMS simulations below 400

K will not be further discussed here.'

In addition, we added a comparison between mean age of air from global 3-dimensional CLaMS simulations and trajectory-based transport times to observation-based mean age of air derived from $N_2O$ measurements to the revised version of the manuscript.

21. *4.2 Fig.6 should have the Tibetan plateau outlined and it should be compared with Fig. 4 of B2020*

We abstain to include political borders to Fig. 6 of Vogel et al. (2023b) to avoid any political discussions. For a detailed comparison between Fig. 4 of Bucci et al. (2020) and Fig. 6 of Vogel et al. (2023b), we refer to Stroh and StratoClim-Team (2023) that will be hopefully soon be available to the public.

22. *4.2 Use a readable color scale for Fig. A2 and add it to the figure*

We added the legend in Fig. A2 of the revised version of the paper. The legend was cropped by accident in Vogel et al. (2023b).

23. *4.2 Fig.7 should be compared with Fig. 10 of B2020*

In Fig. 10 of Bucci et al. (2020) the convective source regions as well as the fraction of air masses recirculating within the Asian monsoon anticyclone depending on altitude are shown for the Stratoclim flights F01-F08 carried out from Kathmandu in 2017. In Bucci et al. (2020) a TRACZILLA back-trajectory analysis is used with a trajectory length of 30 d in contrast to our study going back about one year. The differences between TRACZILLA and CLaMS are described in more detail above (#1). We added the following discussion to Sect. 4.2 of Vogel et al. (2023b).

Using cloud top altitudes from geostationary satellites to identify convection which occurred 30 days before the StratoClim measurements, Bucci et al.

(2020) (see Fig. 10 therein) found that up to an altitude of $17\,\text{km}$ ($\sim 400\,\text{K}$) convective sources contribute more than 95% to the composition of the air probed during all flights. However, they calculate back-trajectories only back to cloud top altitudes. Nevertheless, in CLaMS trajectories only the convection inherent in the ERA5 and ERA-Interim reanalyses is included and therefore, small scale convection could be underestimated. Further, in Bucci et al. (2020) only very young air masses (younger than $30\,\text{d}$) are considered, therefore contributions from pre-monsoon 2017 and winter 16/17 are not covered. A more detailed comparison between the approach used in Bucci et al. (2020) and our analysis can be found elsewhere (Stroh and StratoClim-Team, 2023).

24. *4.2 l.312-314 : "However ..." I do not understand this sentence*

We agree that this sentence is misleading, therefore we removed (L312) in the revised version:

'However the vertical dispersion in ERA5 $1° \times 1°$ is higher as in ERA5 and ERA-Interim (in particular above $440\,\text{K}$) caused by the down-scaling to a $1° \times 1°$ horizontal grid and a 6-hourly time resolution loosing some details of upward transport along the trajectories.'

25. *4.3 l.322 Add that iit is the first day after the beginning of the backward trajectory if it is the case.*

To clarify this point, we revised the sentence as follows;

'The effective ascent rates for $1\,\text{day}$ reflect the short term evolution of an air mass and can be impacted by recent convective events (e.g. Fig. 9b at $390\,\text{K}$) or stratospheric intrusions i.e. mixing with older stratospheric air (Fig. 9b at $\sim 415\,\text{K}$ and $\sim 435\,\text{K}$).'

$\longrightarrow$

The effective ascent rates calculated over 24 h just before the aircraft measurements (1 day) reflect the short term evolution of the sampled air mass and can be impacted by recent convective events (e.g. Fig. 9b at 390 K) or stratospheric intrusions i.e. mixing with older stratospheric air (Fig. 9b at ∼415 K and ∼435 K).

26. *4.3 l.341 : What do you mean by "an idealized parcel" ? Do you mean an age calculation made on a single parcel without any averaging ? Nobody is doing that and this sentence is basically misleading and useless. Please explain better why the two calculatios can be compared.*

We agree and removed the following sentence (L341):

'Therefore, observation-based mean ascent rates are lower limits compared to an idealised air parcel ascending without any mixing with aged stratospheric air.'

27. *4.4 I wonder why this short section and Fig.11 has been dropped here and not merged with 4.1.*

We agree that Sect. 4.4 of Vogel et al. (2023b) is very short and merged it with Sect. 4.3 of Vogel et al. (2023b).

28. *4.4 Fig.11 should be drawn with logarithnmic vertical axis.*

We prefer to use a linear y-axis instead of a logarithmic scale recommended by Reviewer #2. The difference between a linear and logarithmic y-axis is shown in Fig. 1 of this reply.

29. *4.5 This section is not at all about CO2 but purely about transport and ages. Fig. 12 does not display any obvious differences between the three cases and what is new in Fig.13 with respect to Fig. 4. I believe this section is a remain of an early version of this manuscript derived from V2023. The content should be merged with 4.1*

[Figure]

Figure 1: Normalised frequency distribution of the transport time from $360\,\mathrm{K}$ ($\approx$ the level of maximum convective outflow) to the location of the aircraft measurement along the CLaMS backward trajectories (denoted as age spectrum) using ERA-Interim with linear (left) and logarithmic (right) y-axis. The age spectra are shown for different levels of potential temperature (for $2\,\mathrm{K}$ intervals) for a time resolution of 5 and 10 days.

Following the advice of both reviewers, we remove Sect. 4.5 (L382-402 and Fig. 13) in the revised version of the manuscript. Parts of the text included in Sect. 4.5 are revised and shifted to Sect. 3.2 and 4.6 (further details see below). Figure 12 of Vogel et al. (2023b) is revised to better demonstrate the differences in the trajectory-based transport times (see Fig. 2 of this reply.

Airborne $CO_2$ measurements from the StratoClim campaign in Kathmandu (Nepal) during July and August 2017 are shown in Fig. 2 (of this reply). Each air parcel is coloured by the trajectory-based transport time from the model BL to the time of measurements inferred by Lagrangian back-trajectory calculations driven by ERA-Interim (Fig. 2a of this reply). Trajectory-based transport times increase with the altitude of sampled air parcels (as already shown in Fig. 4 of Vogel et al. (2023b)). However, there is also a strong variability of transport times between individual air parcels at the same level of potential temperature indicating mixing of air masses of different transport times or of different age (Fig. 2a of this reply). Moreover, differences in transport times of individual air parcels are found using instead of ERA-Interim ERA5 (Fig. 2b of this reply) reanalyses. In the stratosphere ERA-Interim has the tendency to be faster (shorter transport times) than ERA5 (bluish data points). In the UTLS, certain air masses are found experienced faster upward transport by convection using ERA5 (reddish points).

30. *4.5 l.372-374 The right location for generalities on CO2 is the introduction.*

Many thanks for this comment. We shifted L372-374 to the beginning of Sect. 3.2. making some small adjustments.

'High-resolution $CO_2$ profiles measured in situ (Fig. 12) reflect the seasonal variability of $CO_2$ at ground level (see Fig. 2). $CO_2$ concentrations are relatively independent from diurnal variations in the UTLS, although $CO_2$ has a strong diurnal cycle near the ground. Further, $CO_2$ is chemically inert in the troposphere and stratosphere and can be used as an age tracer considering time periods of several months (e.g. Boering et al., 1996; Andrews et al., 2001; Ray et al., 2022).'

[Figure]

Figure 2: Airborne $CO_2$ measurements from the StratoClim campaign in Kathmandu (Nepal) during July and August 2017. Each air parcel is coloured by the trajectory-based transport time from the model boundary layer (BL) to the time of measurements inferred by Lagrangian back-trajectory calculations driven by ERA-Interim (a). Air parcels located in the model BL are not shown. Aged air (air located in the free atmosphere on 1 June 2016) is marked in black. Further, the differences of the trajectory-based transport time between ERA-Interim and ERA5 (b) are shown from back-trajectories reaching the model BL. In addition, the mean WMO tropopause (Hoffmann and Spang, 2022) as well as the lowest and highest tropopause (grey dashed lines) over Kathmandu during the aircraft campaign (27 July - 10 August 2017) are added using ERA-Interim (a) and ERA5 (b) reanalysis.

$\longrightarrow$

Vogel et al. (2023a) demonstrated that high-resolution $CO_2$ profiles measured in situ during the StratoClim campaign in summer 2017 reflect the seasonal variability of $CO_2$ at ground level. In addition, $CO_2$ is chemically inert in the troposphere and stratosphere and can be used as an age tracer considering time periods of several months (e.g. Boering et al., 1996; Andrews et al., 2001; Ray et al., 2022). Therefore a reasonable reconstruction of vertical $CO_2$ was conducted successfully using CLaMS back-trajectories driven by ERA5 reanalysis using ground-based $CO_2$ measurement (Vogel et al., 2023a).

31. *4.5 l.381 : Add a reference to Tegtmeier et al., (2020) when discussing the tropopause shift between ERA-Interim and ERA5. This shift is mainly due to the higher vertical resolution of the ERA5 with 3 times more levels in the TTL. The improved hozizontal resolution in ERA5 is also inducing the tropopause to jump up and down in convective regions. It is perhaps not meaningful to define a tropopause from WMO criterion at such high resolution. Calculating the tropopause on horizontally filtered data is more reasonable. In prcatice 1° is a good compromise. However, using a 1° subsampling, as I suspect to be the case here, does not removde the problem.*

For better clarity, we revised L381 as follows.

'Hoffmann and Spang (2022) found that the standard deviation of the tropical tropopause height are $\approx$ 30-50% higher in ERA5 compared to ERA-Interim, related to the higher resolution of ERA5.'

$\longrightarrow$

Hoffmann and Spang (2022) found that the standard deviation of the tropical tropopause height are $\approx$ 30-50% higher in ERA5 ($T_L$ 639; high-resolution version) compared to ERA-Interim, mostly related to explicitly resolved gravity waves in ERA5, which are absent in ERA-Interim due to its coarser spatial resolution. Tegtmeier et al. (2020) attributed tropopause shifts be-

tween ERA-Interim and ERA5 to higher vertical resolution of ERA5, having 3 times more levels in the tropical tropopause than ERA-Interim.

32. *4.5 Defining the level of maximum CO2 in the TTL is the only usage of CO2 here. The sentence on l.392-394 is badly written but also wrong. The transport properties from the ground have not been uniform over the range of time from the pre-monsoon to the beginning of August and the CO2 profile is a convolution between the source variation and the changing transport. This is exactly what is done in the CO2 reconstruction discussed in 4.6 and therefore this naive interpretation is totally at odd.*

   As stated above (#29), we removed L392-394 to avoid any misunderstanding.

33. *4.5 l.400. Is it not true that the reconstruction up to 400 K is due to NTL and so, what it doing CL here.*

   As stated above (#29), we removed L400 to avoid any misunderstanding.

34. *4.6 It is not clear from the reconstruction shown in the bottom row of Fig. 14 that ERA5 should be preferred to ERA-Interim. Both are good below 400 K but above this level ERA-Interim is better and with less dispersion. The reconstruction below 400 K is based on young trajectories and on the CO2 cycle over north India while the reconstruction above 400 K depends on a much wider distribution of sources, both in time and in space. The result is somewhat infortunate and of course cannot be used to support a general conclusion that ERA5 is better but thre were perhaps here too much problems to solve at once.*

   Many thanks for this comment, we agree that the difference between a $CO_2$ reconstruction using ERA5 and ERA-Interim is small, therefore we revised L434 as follows.

   'However, a $CO_2$ reconstruction using ERA5 agrees best with the vertical measured $CO_2$ profile up to 410 K.'

$\longrightarrow$

A $CO_2$ reconstruction using ERA5 agrees best with the vertical measured $CO_2$ profile up to 410 K, although there is only a slight difference when using Era-Interim reanalysis. It should be noted that the used $CO_2$ reconstruction technique has limitations because of the very low number of sites measuring ground-based CO2 over the Indian subcontinent in 2016-2107.

35. *4.6 In both rows of Fig.14, there is a sort of jump at 400K. Is it a feature or a technical issue ?*

Yes, there is a discontinuity at about 400 K. Below 400 K, young air masses released during the monsoon 2017 dominate. However, between 400 and 420 K, air masses from pre-monsoon 2017 and winter 16/17 have to be taken into account originating mainly in the southern and northern ITCZ with partly extreme low $CO_2$ values (e.g. Samoa and Warm Pool region). Thus at these altitudes a change in the origins as well as in the age of the air masses occurs. In addition, from 400 K to 450 K we have a much lower number of back-trajectories along the research flights according the flight tracks of flights F01-F08 (see Figure A3 in Vogel et al. (2023b)) compared to other flight altitudes and therefore a lower statistic. To illustrate that even better, we add to this reply Fig. 3 (a previous version of Fig. 4 of Vogel et al. (2023b)) showing the number of trajectories depending from level of potential temperature.

**References**

Andrews, A. E., Boering, K. A., Daube, B. C., Wofsy, S. C., Loewenstein, M., H., Podolske, J. R., Webster, C. R., Herman, R. L., Scott, D. C., Flesch, G. J., Moyer, E. J., Elkins, J. W., Dutton, G. S., Hurst, D. F., Moore, F. L., Ray, E. A., Romashkin, P. A., and Strahan, S. E.: Mean age of stratospheric air derived from in situ observations of $CO_2$, $CH_4$ and $N_2O$, J. Geophys. Res., 106, 32 295–32 314, 2001.

Boering, K. A., Wofsy, S. C., Daube, B. C., Schneider, H. R., Loewenstein, M.,

[Figure]

Figure 3: A previous version of Fig. 4 of Vogel et al. (2023b) only for ERA5 data, but showing in addition the number of trajectories depending from level of potential temperature.

Podolske, J. R., and Conway, T. J.: Stratospheric Mean Ages and transport rates from observations of carbon dioxide and nitrous oxide, Science, 274, 1340–1343, 1996.

Bucci, S., Legras, B., Sellitto, P., D'Amato, F., Viciani, S., Montori, A., Chiarugi, A., Ravegnani, F., Ulanovsky, A., Cairo, F., and Stroh, F.: Deep-convective influence on the upper troposphere–lower stratosphere composition in the Asian monsoon anticyclone region: 2017 StratoClim campaign results, Atmos. Chem. Phys., 20, 12 193–12 210, https://doi.org/10.5194/acp-20-12193-2020, 2020.

Clemens, J., Hoffmann, L., Vogel, B., Griessbach, S., and Thomas, N.: Implementation of diabatic advection in the Lagrangian transport model MPTRAC 3.0: A comparative evaluation within a hierarchy of transport uncertainties, Geosci. Model Dev., to be submitted, 2023a.

Clemens, J., Vogel, B., Hoffmann, L., Griessbach, S., Thomas, N., Fadnavis, S., Müller, R., Peter, T., and Ploeger, F.: Identification of source regions of the Asian Tropopause Aerosol Layer on the Indian subcontinent in August

2016, EGUsphere, 2023, 1–39, https://doi.org/10.5194/egusphere-2022-1462, accepted, 2023b.

Diallo, M., Legras, B., and Chédin, A.: Age of stratospheric air in the ERA-Interim, Atmos. Chem. Phys., 12, 12 133–12 154, https://doi.org/10.5194/acp-12-12133-2012, 2012.

Fadnavis, S., Ravi Kumar, K., Tiwari, Y. K., and Pozzoli, L.: Atmospheric $CO_2$ source and sink patterns over the Indian region, Annales Geophysicae, 34, 279–291, https://doi.org/10.5194/angeo-34-279-2016, 2016.

Hoffmann, L. and Spang, R.: An assessment of tropopause characteristics of the ERA5 and ERA-Interim meteorological reanalyses, Atmos. Chem. Phys., 22, 4019–4046, https://doi.org/10.5194/acp-22-4019-2022, 2022.

Hoffmann, L., Günther, G., Li, D., Stein, O., Wu, X., Griessbach, S., Heng, Y., Konopka, P., Müller, R., Vogel, B., and Wright, J. S.: From ERA-Interim to ERA5: the considerable impact of ECMWF's next-generation reanalysis on Lagrangian transport simulations, Atmos. Chem. Phys., 19, 3097–3124, https://doi.org/10.5194/acp-19-3097-2019, 2019.

Konopka, P., Tao, M., Ploeger, F., Diallo, M., and Riese, M.: Tropospheric mixing and parametrization of unresolved convective updrafts as implemented in the Chemical Lagrangian Model of the Stratosphere (CLaMS v2.0), Geosci. Model Dev., 12, 2441–2462, https://doi.org/10.5194/gmd-12-2441-2019, 2019.

Konopka, P., Tao, M., von Hobe, M., Hoffmann, L., Kloss, C., Ravegnani, F., Volk, C. M., Lauther, V., Zahn, A., Hoor, P., and Ploeger, F.: Tropospheric transport and unresolved convection: numerical experiments with CLaMS-2.0/MESSy, Geosci. Model Dev., 15, 7471–7487, https://doi.org/10.5194/gmd-15-7471-2022, 2022.

Legras, B. and Bucci, S.: Confinement of air in the Asian monsoon anticyclone and pathways of convective air to the stratosphere during the summer season, Atmos. Chem. Phys., 20, 11 045–11 064, https://doi.org/10.5194/acp-20-11045-2020, 2020.

Li, D., Vogel, B., Müller, R., Bian, J., Günther, G., Ploeger, F., Li, Q., Zhang, J., Bai, Z., Vömel, H., and Riese, M.: Dehydration and low ozone

in the tropopause layer over the Asian monsoon caused by tropical cyclones: Lagrangian transport calculations using ERA-Interim and ERA5 reanalysis data, Atmos. Chem. Phys., 20, 4133–4152, https://doi.org/10.5194/acp-20-4133-2020, 2020.

Li, F., Waugh, D. W., Douglass, A. R., Newman, P. A., Pawson, S., Stolarski, R. S., Strahan, S. E., and Nielsen, J. E.: Seasonal variations in stratospheric age spectra in GEOSCCM, J. Geophys. Res., 117, D5, https://doi.org/10.1029/2011JD016877, 2012.

Mahowald, N. M., Plumb, R. A., Rasch, P. J., del Corral, J., and Sassi, F.: Stratospheric transport in a three-dimensional isentropic coordinate model, J. Geophys. Res., 107, 4254, https://doi.org/10.1029/2001JD001313, 2002.

Nomura, S., Naja, M., Ahmed, M. K., Mukai, H., Terao, Y., Machida, T., Sasakawa, M., and Patra, P. K.: Measurement report: Regional characteristics of seasonal and long-term variations in greenhouse gases at Nainital, India, and Comilla, Bangladesh, Atmos. Chem. Phys., 21, 16 427–16 452, https://doi.org/10.5194/acp-21-16427-2021, 2021.

Ploeger, F. and Birner, T.: Seasonal and inter-annual variability of lower stratospheric age of air spectra, Atmos. Chem. Phys., 2016, 10 195–10 213, https://doi.org/10.5194/acp-16-10195-2016, 2016.

Ploeger, F., Diallo, M., Charlesworth, E., Konopka, P., Legras, B., Laube, J. C., Grooß, J.-U., Günther, G., Engel, A., and Riese, M.: The stratospheric Brewer–Dobson circulation inferred from age of air in the ERA5 reanalysis, Atmos. Chem. Phys., 21, 8393–8412, https://doi.org/10.5194/acp-21-8393-2021, 2021.

Pommrich, R., Müller, R., Grooß, J.-U., Konopka, P., Ploeger, F., Vogel, B., Tao, M., Hoppe, C. M., Günther, G., Spelten, N., Hoffmann, L., Pumphrey, H.-C., Viciani, S., D'Amato, F., Volk, C. M., Hoor, P., Schlager, H., and Riese, M.: Tropical troposphere to stratosphere transport of carbon monoxide and long-lived trace species in the Chemical Lagrangian Model of the Stratosphere (CLaMS), Geosci. Model Dev., 7, 2895–2916, https://doi.org/10.5194/gmd-7-2895-2014, 2014.

Ray, E. A., Atlas, E. L., Schauffler, S., Chelpon, S., Pan, L., Bönisch, H., and Rosenlof, K. H.: Age spectra and other transport diagnostics in the North American monsoon UTLS from SEAC[4]RS in situ trace gas measurements, Atmos. Chem. Phys., 22, 6539–6558, https://doi.org/10.5194/acp-22-6539-2022, 2022.

Stroh, F. and StratoClim-Team: First detailed airborne and balloon measurements of microphysical, dynamical, and chemical processes in the Asian Summer Monsoon Anticyclone: overview and selected results of the 2016/2017 StratoClim field campaigns, Atmos. Chem. Phys., to be submitted, 2023.

Tegtmeier, S., Anstey, J., Davis, S., Dragani, R., Harada, Y., Ivanciu, I., Pilch Kedzierski, R., Krüger, K., Legras, B., Long, C., Wang, J. S., Wargan, K., and Wright, J. S.: Temperature and tropopause characteristics from reanalyses data in the tropical tropopause layer, Atmos. Chem. Phys., 20, 753–770, https://doi.org/10.5194/acp-20-753-2020, 2020.

Vogel, B., Volk, C. M., Wintel, J., Lauther, V., Müller, R., Patra, P. K., Riese, M., Terao, Y., and Stroh, F.: Reconstructing high-resolution in-situ vertical carbon dioxide profiles in the sparsely monitored Asian monsoon region, Commun Earth Environ, 4, https://doi.org/10.1038/s43247-023-00725-5, 2023a.

Vogel, B., Volk, M., Wintel, J., Lauther, V., Clemens, J., Grooß, J.-U., Günther, G., Hoffmann, L., Laube, J. C., Müller, R., Ploeger, F., and Stroh, F.: Evaluation of vertical transport in the Asian monsoon 2017 from $CO_2$ reconstruction in the ERA5 and ERA-Interim reanalysis, EGUsphere, 2023, 1–37, https://doi.org/10.5194/egusphere-2023-1026, 2023b.